# Pyruvate dehydrogenase operates as an intramolecular nitroxyl generator during macrophage metabolic reprogramming

Erika M. Palmieri [1], Ronald Holewinski[2], Christopher L. McGinity[1], Ciro L. Pierri[3], Nunziata Maio [4], Jonathan M. Weiss [1], Vincenzo Tragni[3], Katrina M. Miranda [5], Tracey A. Rouault[4], Thorkell Andresson[2], David A. Wink[1] & Daniel W. McVicar [1] ✉

M1 macrophages enter a glycolytic state when endogenous nitric oxide (NO) reprograms mitochondrial metabolism by limiting aconitase 2 and pyruvate dehydrogenase (PDH) activity. Here, we provide evidence that NO targets the PDH complex by using lipoate to generate nitroxyl (HNO). PDH E2-associated lipoate is modified in NO-rich macrophages while the PDH E3 enzyme, also known as dihydrolipoamide dehydrogenase (DLD), is irreversibly inhibited. Mechanistically, we show that lipoate facilitates NO-mediated production of HNO, which interacts with thiols forming irreversible modifications including sulfinamide. In addition, we reveal a macrophage signature of proteins with reduction-resistant modifications, including in DLD, and identify potential HNO targets. Consistently, DLD enzyme is modified in an HNO-dependent manner at Cys[477] and Cys[484], and molecular modeling and mutagenesis show these modifications impair the formation of DLD homodimers. In conclusion, our work demonstrates that HNO is produced physiologically. Moreover, the production of HNO is dependent on the lipoate-rich PDH complex facilitating irreversible modifications that are critical to NO-dependent metabolic rewiring.

Macrophages undergo profound metabolic changes that support their function in health and disease. The combination of lipopolysaccharide (LPS) and interferon-γ leads to differentiation of pro-inflammatory macrophages, often referred to as "M1" cells, with enhanced glycolytic flux and glutamine intake, and substantially reduced oxidative phosphorylation[1]. Glycolysis supplies the pentose phosphate pathway supporting the generation of NADPH to fuel the NADPH-oxidases for pathogen killing, as well biosynthetic pathways through the production of ribose[2]. Concurrently, nitric oxide (NO) following activation is directly lethal to microorganisms, while supporting glycolytic commitment by shutting down the use of pyruvate in the TCA cycle[3–5]. In addition, NO suppresses mitochondrial aconitase (ACO2) activity, leading to the accumulation of citrate, which upon export to the cytoplasm supports inflammatory lipid production[6–8]. Lastly, upregulation of aconitate decarboxylase (Acod)−1 leads to the production of the antimicrobial metabolite itaconate from the TCA intermediate cis-aconitate until NO concentration reaches levels that inhibit ACO2 function[6,9]. Given that

[1]Cancer Innovation Laboratory, NCI-Frederick, Frederick, MD 21702, USA. [2]Protein Characterization Laboratory, Frederick National Laboratory for Cancer Research, Leidos Biomedical Research, Inc., Frederick, MD 21702, USA. [3]Laboratory of Biochemistry, Molecular and Structural Biology, Department of Pharmacy-Pharmaceutical Sciences, University of Bari, Via E. Orabona, 4, Bari 70125, Italy. [4]Molecular Medicine Branch, Eunice Kennedy Shriver National Institute of Child Health and Human Development, 9000 Rockville Pike, Bethesda, MD 20892, USA. [5]Department of Chemistry and Biochemistry, University of Arizona, Tucson, AZ 85721, USA. ✉e-mail: mcvicard@mail.nih.gov

various metabolites such as NO drive unique phenotypic and metabolic characteristics[8,10], more study is required to investigate closely the molecular mechanisms driving metabolic adaptation during stimulation.

A critical, but less studied, component of this metabolic reorganization is the inhibition of the pyruvate dehydrogenase (PDH) complex slowing glucose-derived carbon from entering the TCA cycle[11,12]. Due to the relatively rapid kinetics of NOS2-dependent PDH shutdown during macrophage polarization, as well as the apparent lack of involvement of HIF-1α in enzyme inhibition[6], we hypothesized that NO might directly target components of the PDH complex.

The PDH complex consists of three different enzymes, E1, E2, and E3, which catalyze the conversion of pyruvate to acetyl-Coenzyme A (CoA), yielding NADH. Lipoate covalently bound to E2 is fundamental in accepting the acetyl group from E1, and transferring it to reduced CoA[13]. The resulting dihydrolipoate undergoes oxidation by the E3 enzyme (also referred to as dihydrolipoamide dehydrogenase; DLD) producing NADH. Interestingly, the DLD enzyme functions as a homodimer not only in PDH but also in α-ketoglutarate dehydrogenase (OGDH), branched-chain amino acid-dehydrogenase (BCKDH) and the glycine cleavage system (GCS), all within the mitochondrial matrix[14,15].

Over the last 30 years, substantial attention has been given to then biological effects of NO and related oxidized nitrogen oxides such as $NO_2$ and $N_2O_3$. Nitroxyl (HNO) is more reduced than NO, and while the possibility of endogenous formation has long been hypothesized[16–20], the mechanisms have yet to be definitively established. Plausible routes for endogenous HNO production include the reaction of NO with hydrogen sulfide ($H_2S$) and nucleophilic attack of an S-nitrosothiol (RSNO) by an adjacent thiol[17,21–27].

Compared with NO and other reactive nitrogen species (RNS), HNO has higher electrophilicity[28], a distinct mechanism of action[20,29,30] and high reaction rates with critical targets including thiols[29,31]. Modification of thiols by HNO, to produce a reduction-resistant, irreversible sulfinamide ($RS(O)NH_2$) has been observed in a variety of cellular systems[17,28]. In part due to its ability to mediate reduction-resistant modifications driving enzyme inhibition[32], there has been substantial interest in a physiological role for HNO as an alternative signaling agent to NO. HNO donors induce vasodilation and contribute to the prevention of atherosclerosis and vascular thrombosis, and HNO is a promising alternative clinical cardiac inotropic agent in heart failure[33–41]. HNO donors also show efficacy in the treatment of alcoholism, cancer[42] and pain[43,44]. In sum, the distinct chemistry of HNO compared to NO suggest that biological targeting by HNO may be selective and localized, with critical thiols being hot spots or privileged targets[45]. As such, HNO has high potential to function as a tightly controlled signaling molecule, which can be exploited therapeutically[19,32,39].

Although there are numerous chemical processes by which HNO could be generated physiologically, and ample evidence of its effects, until now endogenous HNO and its biochemical fingerprint has yet to be fully established, perhaps in part due to it propensity for self-reaction[28]. Here, we investigate the mechanism of inhibition of PDH and identify a model of intramolecular generation of HNO during macrophage M1 polarization. This mechanism involves NO attack of PDH-E2 associated lipoates. The unique thiol configuration of lipoate facilitates the subsequent generation of HNO in close proximity with PDH-distal DLD. Irreversible, HNO-derived modification of DLD results in potent inhibition of PDH and mitochondrial adaptation to stimulation. This mechanism is consistent with the parallel inhibition of other DLD-containing enzyme complexes including OGDH and BCKDH. Intramolecular HNO generation in biological settings may provide a mechanism to induce site-specific responses while preventing uncontrolled modifications.

## Results

### Metabolic reprogramming of macrophages at PDH is dependent on NO fluxes and is associated with alterations of PDH-E2 cofactor lipoate

We previously showed that metabolic adaptation due to PDH inhibition is NOS2-dependent[6] and we sought to examine this mechanism in more detail. Using bone-marrow derived macrophages (BMDM), we first measured PDH enzymatic activity and oxygen consumption rates (OCR) of permeabilized cells fueled with pyruvate; both decrease in an NO-dependent manner (Fig. 1a, b). As lipoate levels have been shown by us and others to be altered during macrophage activation[46,47], we supplemented cell cultures with lipoate and found that this ameliorated reduction of OCR by NO (Fig. 1c). Since NO-derived RNS such as $N_2O_3$ interact with thiol groups, we considered that NO might target PDH through its E2-bound cofactor, lipoate (Fig. 1d). To test this possibility, we performed high resolution proteomics on the E2 enzyme of BMDMs (Fig. S1a). We detected lipoyl-bound peptides containing Lys[131] and Lys[258] of E2 (Fig. 1e) and higher abundance and lipoylation rates for Lys[131] compared to Lys[258] was observed (Fig. S1b–e). When NO was present, our analysis detected a form of lipoate consistent with S-nitrosation on one thiol and acetylation of the second thiol of the same molecule ($K-C_{10}H_{16}O_3S_2N$) (Fig. 1f). Although less abundant than on Lys[131], this acetyl-nitroso lipoate modification was also found on Lys[258] (Fig. S1f). We did not detect substantial signal consistent with a mixed S-nitroso/reduced thiol (Fig. S1g) or a dual S-nitroso modification on lipoate (Fig. S1h). Due in part to steric interactions, the latter case is not expected, while S-nitrosation of thiols is commonplace[48]. The apparent lack of a mixed S-nitroso/reduced thiol lipoate, ($K-C_8H_{15}O_3S_2N$), compatible also with a lipoyl-sulfinamide or a lipoyl-hydroxyl-disulfenamide, is consistent with increased reactivity. In particular, when thiol/-SNO ratios are low such as when protein -SNOs are accessible only to other proximal protein thiols, HNO generation is a possibility[17]. Moreover, in the presence of NO, BMDM peptides accumulated that contain lysine bound to dithiol- and acetylated-lipoate (Fig. 1g, h). This suggests that the enzyme is inhibited in this configuration most likely because of lack of cofactor recycling. Thus, the S-nitroso-acetyl-lipoate modification likely represents S-nitrosation occurring after the enzyme is inhibited at the acetylated state.

### Modifications on DLD and the E3 binding protein of PDH are compatible with HNO exposure

We previously demonstrated a decline in DLD activity in macrophages stimulated with LPS + IFNγ that is dependent on NOS2 and occurs concurrently with altered migration in native gels[6], suggesting direct modification of DLD by NO. This raises the possibility that other mitochondrial dehydrogenases sharing DLD (Fig. S2a) would be similarly targeted in macrophages. Metabolomics from stimulated WT and *Nos2*[−/−] macrophages suggest that BCKDH, OGDH and GCS are indeed targeted in a NO-dependent manner (Fig. S2b). To further strengthen this conclusion, we performed isotope tracing analysis using [U-¹³C6] Leucine to assess BCKDH activity (Fig. 2a). A 4-h pulse with [U-¹³C6] Leucine resulted in >70% of intracellular leucine being labeled. Approximately 40% of α-ketoisocaproate was also labeled suggesting prompt transamination, with equal rates of fractional labeling between WT and *Nos2*[−/−] that was independent of stimulation status (Fig. S2c). As anticipated, we observed decreased [U-¹³C6]Leucine-derived carbon incorporation downstream of BCKDH in stimulated WT BMDMs but not in stimulated *Nos2*[−/−] cells (Figs. 2b, c and S2d, e). The isotopologue incorporation pattern confirmed predominant changes in the M + 5 and M + 3 isotopologues of 3-hydroxy-3-methylglutarate (HMG) and acetoacetate, respectively, indicative of disruption of BCKDH activity in a NOS2-dependent manner (Fig. 2b, c).

NO was sufficient to inhibit DLD activity (Fig. 2d), and treating intact cells with lipoate restored DLD functionality (Fig. S2f),

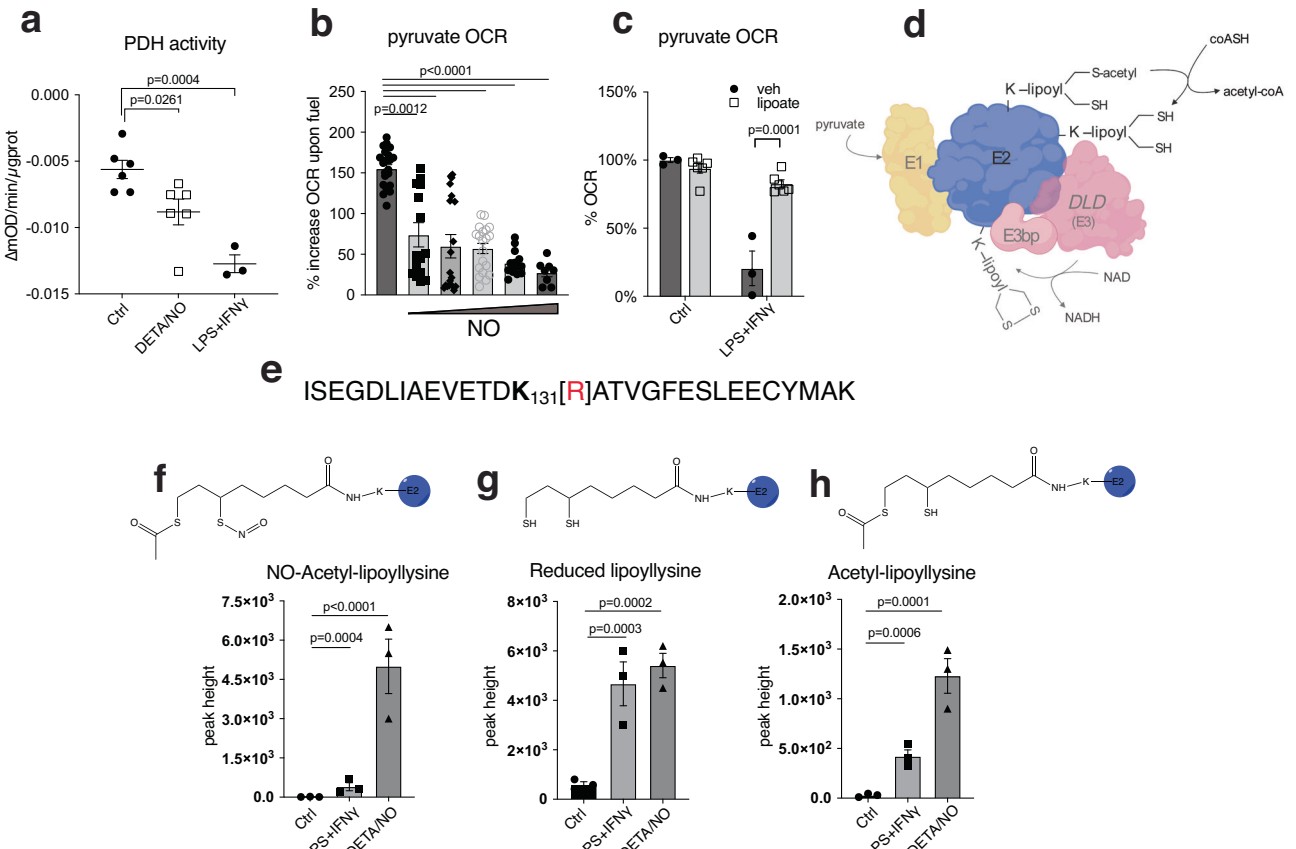

**e** ISEGDLIAEVETDK$_{131}$[R]ATVGFESLEECYMAK

**Fig. 1 | NO fluxes regulate PDH and alter PDH-E2 cofactor lipoate in macrophages. a** Enzymatic activity of PDH in total cell lysates from WT BMDMs from male and female mice (mixed sex in all groups) after 16 h activation with LPS + IFNγ or DETA/NO (100 µM) ($n > 3$ mice). **b** Representative Seahorse analysis of permeabilized WT BMDMs from male and female mice (mixed sex in all groups) where state 3-OCR was elicited by a mixture of pyruvate and malate to measure PDH flux. Bar graphs show quantified respiration in 16 h activated WT ($n > 8$ wells). Data in (**a**) and (**b**) were analyzed by one-way ANOVA with Dunnet's multiple comparisons test. **c** Quantified pyruvate/malate respiration from Seahorse analysis of permeabilized WT BMDMs from male and female mice (mixed sex in all groups) after 8 h activation, performed in the presence of vehicle or lipoic acid ($n > 3$ mice) shown as percentage of OCR relative to untreated. Data were analyzed by two-way ANOVA (Sidak's post-tests). **d** Schematic illustration of PDH enzyme with emphasis on E2 subunit. **e** Interrogated peptide of PDH-E2 predicted to bind lypoyl group on functional lysine (K). **f**–**h** Major products, possible structures and abundances of E2 K$^{131}$-lipoylated peptide in mitochondrial lysates from WT BMDMs from male mice after 16 h stimulation with LPS + IFNγ or DETA/NO (500 µM) ($n = 3$ technical repeats) Data were analyzed by two-way ANOVA with Dunnet's multiple comparisons test. All error bars display mean ± SEM. Data shown are representative of two or more independent experiments. Source data are provided as a Source Data file.

consistent with our OCR results (Fig. 1c). However, adding lipoate to lysates did not restore DLD diaphorase activity (Fig. 2d), suggesting its actions are not due to reduction. Accordingly, treatment with DTT was also not effective in restoring DLD activity, indicating non-reversible modification (Fig. 2d). Exogenous lipoate caused no substantial changes in GSSG/GSH levels but led to an increase in lipoyl-GMP, suggesting activation of lipoate consistent with use of the salvage pathway for protein lipoylation (Fig. S2g, h)[49,50]. Altogether, these data suggest that rather than acting as a reducing agent, exogenous lipoate is likely supporting lipoylation of newly formed E2.

Having determined that NO modifies lipoate, and that lipoate adducts regulate DLD in association with loss of activity, we investigated the chemistry responsible for DLD modification. The reactivity was compatible with formation of a sulfinamide by HNO formed through nucleophilic attack of a lipoyl-SNO by the vicinal thiol (Fig. 2e). Reaction of HNO with a thiol leads to an intermediate that can spontaneously rearrange to a sulfinamide or in the presence of excess thiol can react to produce a disulfide and hydroxylamine (Fig. 2f)[28]. Generation of a sulfinamide in a biological system represents a nearly irreversible thiol modification[51]. Treatment with Angeli's salt (AS), an HNO donor, was sufficient to inhibit DLD activity in BMDMs (Fig. 2g, h) and confirmed the propensity for HNO self-reactivity at high concentrations, where its effect on PDH

is lost. Moreover, compared with an NO donor (DEA/NO) with a comparable half-life of release to AS, AS was a more effective inhibitor in culture (Fig. 2h). Similarly, OGDH activity is preferentially inhibited by HNO as compared to NO (Fig. S2i, j). Lastly, in experiments of potential rescue with DTT, glutathione (GSH) or Cysteine (Cys) in lysates[52], these reducing agents did not substantially restore DLD activity in NO-rich cells (Fig. 2i). In addition, migration of the lipoate containing-E3-binding protein (E3bp) of PDH[53] (Fig. 2j) on native gels changes during stimulation or treatment with AS, but not NO, and this migration is not affected by DTT (Figs. 2k and S2k, l). These data suggest that HNO may be responsible for lipoate-related biochemical alterations in these physiological settings, and that components of PDH and other cellular dehydrogenases may be both HNO producers and targets of inhibition by HNO.

## HNO is generated by NO in the presence of reduced lipoate and is detectable in BMDMs

Given the possible involvement of lipoate groups in endogenous HNO generation, we used chemical approaches to assess if the interaction of lipoate with NO can generate HNO in vitro. Although cellular dithiols can react with S-nitrosoglutathione (GSNO)[54] to generate HNO, lipoate-mediated generation of HNO from NO and thiols has not been shown. We treated GSH with AS or DEA/NO in the presence of reduced or

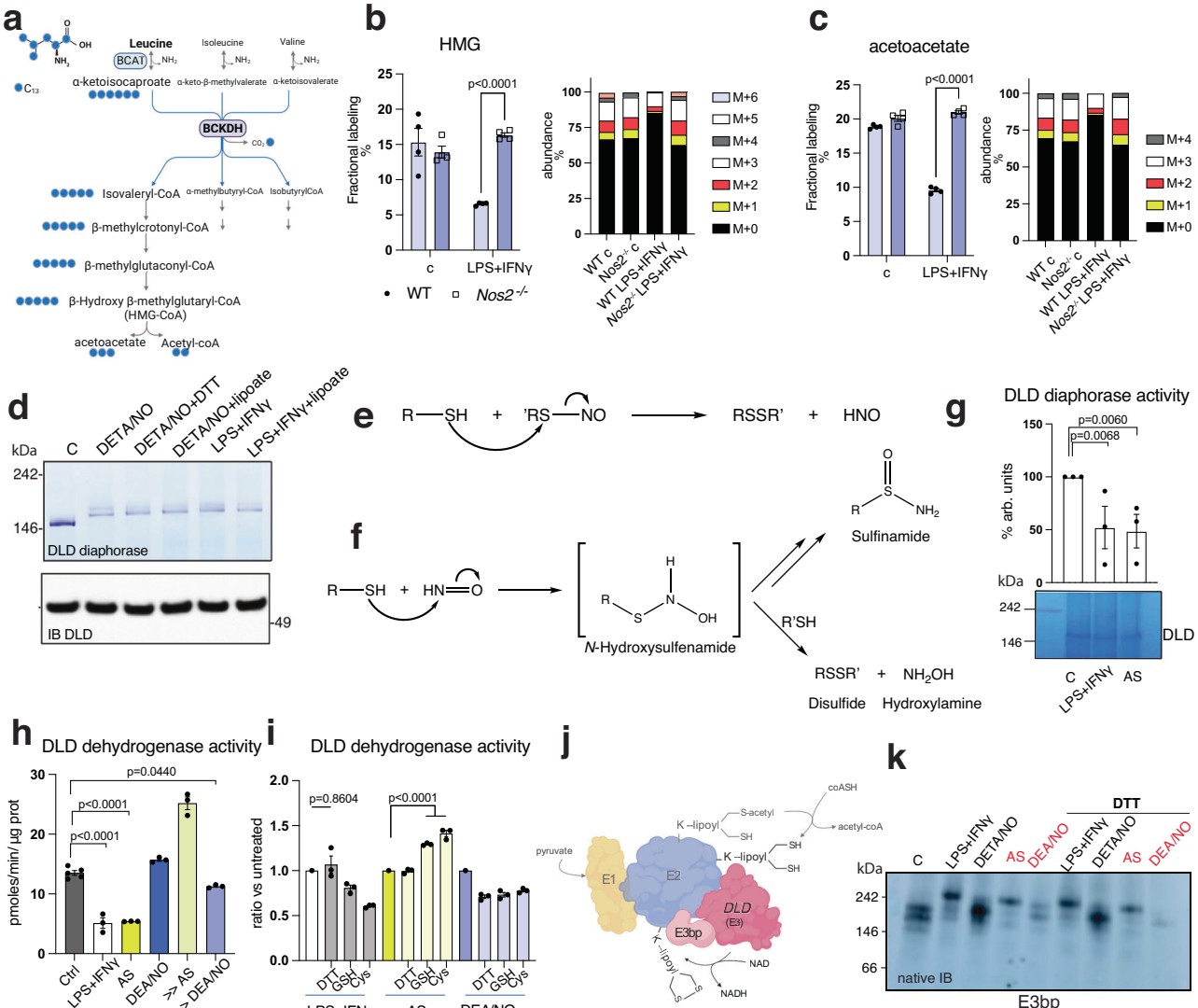

**Fig. 2 | DLD and the E3 binding protein of PDH are likely modified by HNO.**
**a** Schematic illustration of atom transitions using uniformly labeled 13C-Leucine ([U-13C]) (labeled carbons in blue) as tracer for determination of mass isotopologue distributions to infer relative intracellular fluxes. BCAT Branched-chain amino acid aminotransferase, BCKDH Branched-chain α-keto acid dehydrogenase. **b**, **c** WT and $Nos2^{-/-}$ BMDMs from male and female mice (mixed sex in all groups) were activated for 16 h with LPS + IFNγ and cultured 4 h with labeled tracer. Bars show carbon incorporation into M + 0–6 isotopologues of indicated intermediates ($n = 4$ mice per genotype). Data were analyzed by two-way ANOVA with Sidak's post-tests. **d** DLD in-gel activity and native IB of mitochondrial fractions from WT BMDMs untreated or 16 h-stimulated with LPS + IFNγ ($n = 9$ total mice). Treatments with DETA/NO/DTT/lipoate were performed post harvest on lysates. **e** Mechanism of generation of HNO. **f** Outputs of reaction of HNO with common thiols. **g** DLD in-gel activity of mitochondrial fractions from WT BMDMs stimulated for 16 h with

LPS + IFNγ and 4 h with AS. Bar graphs show quantified diaphorase activity ($n = 3$ technical repeats). **h** DLD dehydrogenase activity in mitochondrial lysates from WT after 16 h-stimulation with LPS + IFNγ or either 400μM or 5 mM AS or DEA/NO for 4 h ($n > 3$ mice). **i** Lysates from (**h**) were incubated 1 h at 37 °C with reducing agents (10 mM) before assessment of DLD dehydrogenase activity. Basal values of activity of LPS + IFNγ, AS and DEA/NO from (**h**) were set to 1 ($n > 3$ mice). Data in (**g**–**i**) were analyzed by one-way ANOVA with Dunnet's multiple comparisons test. **j** Schematic illustration of PDH enzyme with emphasis on DLD and E3bp. **k** E3bp-IB on native gels of mitochondrial fractions from BMDMs after 16 h stimulation with LPS + IFNγ or DETA/NO (500 μM), or with AS or DEA/NO for 4 h. Lysates pre-native PAGE were incubated with 10 mM DTT for 1 h at 37 °C ($n = 8$ total mice). All error bars display mean ± SEM. Data shown are representative of two or more independent experiments. $p$ values > 0.05 are reported as "ns" unless otherwise specified. Source data are provided as a Source Data file.

oxidized lipoate and GSH modifications were analyzed by ESI-LC/MS-MS (Fig. S3a). In this system GSH functions as an HNO trap and GSH-sulfinamide (GS(O)NH₂) is a selective biomarker for exposure to HNO[17,29,51,55]. As expected GSH-sulfinamide was detected in the presence of AS (Fig. 3a). Remarkably, we measured substantial GSH-sulfinamide after incubation with DEA/NO and reduced lipoate but not oxidized lipoate (Fig. 3a), highly suggestive that exposure of reduced lipoate to NO can generate free HNO. We also assessed GSH modifications in reactions containing human recombinant DLD (rhDLD) and found that GSH-sulfinamide and sulfinic acid (-SO₂H), a downstream hydrolyzed product, could be further increased (Figs. 3b, c and

S3c) with GSH concentrations dictating the amount of HNO-derived sulfinamide.

Next, we evaluated whether HNO was produced endogenously in our cell system. We incubated murine BMDM with a fluorescent probe for HNO quantification[56] and readily detected an increase in fluorescence in LPS + IFNγ-treated vs unstimulated cells that was probe-specific (Figs. 3d and S3d) and ablated with pre-treatments with the NOS2 inhibitor aminoguanidine (AG). Lastly, to interrogate the requirement of lipoate groups in endogenous HNO generation, we used ML226, an inhibitor of ABHD11, a mitochondrial hydrolase responsible for maintaining correct lipoylation of major α-keto-acid

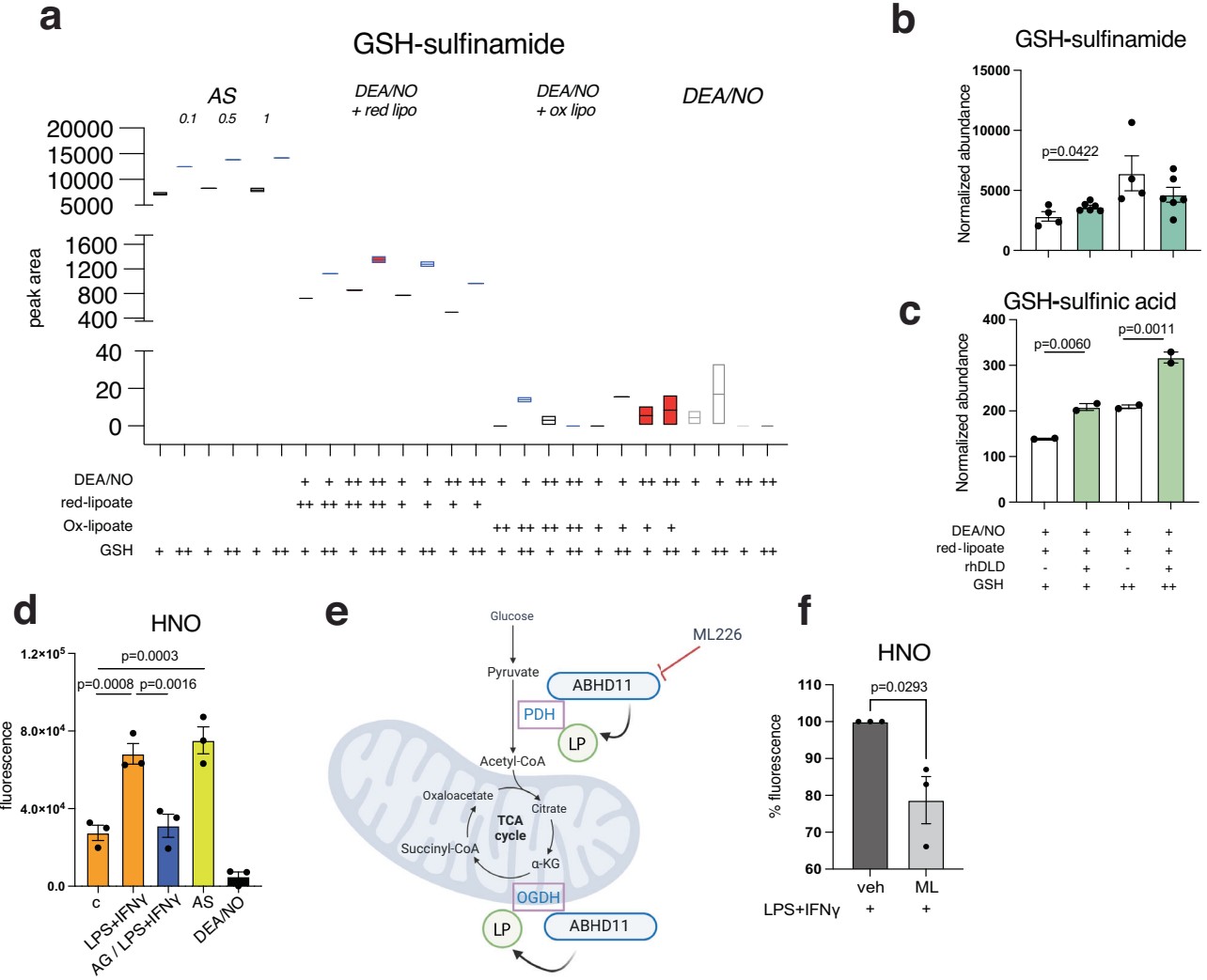

**Fig. 3 | HNO is generated by NO in the presence of reduced lipoate and is detectable in BMDMs. a** Test tube reactions were carried out in the presence of GSH and AS or mixtures of either oxidized or reduced-lipoate with DEA/NO for 1 h at 37 °C. GSH-sulfinamide was quantified by ESI-LC/MS-MS. Boxplots with floating bars (min to max) are shown with line at median. In (**b**) and (**c**) recombinant human DLD (rhDLD) was added to test tube reactions in the presence or absence of reduced lipoate and GSH-sulfinamide and sulfinic acid were quantified ($n > 2$ wells). Data were analyzed by one-way ANOVA with Dunnet's multiple comparisons test. **d** WT BMDMs from male and female mice (mixed sex in all groups) were incubated with fluorescence probe for HNO for 30 min and stimulated with LPS + IFNγ for 4 h or with NOS2 inhibitor Aminoguanidine (AG) 1 h prior to stimulation. After PBS washing, fluorescence was measured in total cell lysates. For AS and DEA/NO conditions, cells were treated with donors for only 30 min. Data ($n = 3$ mice) were analyzed by one-way ANOVA with Dunnet's multiple comparisons test. **e** Schematic illustration of mechanism of action of ML226 on ABHD11, which ensures stability of lipoate groups of PDH and OGDH enzymes. **f** Quantifications (percentage relative to LPS + IFNγ-stimulated cells) of fluorescence for HNO detection in cells treated with ML226 1 h prior to stimulation. Data ($n = 3$ mice) were analyzed by unpaired t test (two-tailed). All error bars display mean ± SEM. Data shown are representative of 2 or more independent experiments except (**b**) which is cumulative data from 2 independent experiments. Source data are provided as a Source Data file.

dehydrogenases[57] (Fig. 3e). ML226 treatment partially inhibited HNO-specific fluorescence (Fig. 3f).

Taken together, these data demonstrate the ability of lipoate to generate HNO, that can be facilitated with additional DLD, and confirm that HNO generation within macrophages is NOS2- and lipoate-dependent, and capable of inducing reduction resistant modifications.

### Proteomic analysis reveals stable Cys-modifications in M1 BMDMs

Next, we sought to detect the effects of endogenous HNO in BMDMs by surveying reduction resistant modified proteins at the level of Cys residues in the mitochondrial-enriched proteome. We undertook an approach with TCEP reduction followed by alkylation and tandem mass tag (TMT) proteomics analysis (Fig. 4a). The rationale included the possibility of detecting sulfinamides, sulfinic acid and sulfonic acid

($-SO_3H$), which, unlike traditional -SNO modifications would be stable during reduction. Moreover, a decrease in alkylation-derived carba-midomethyl (CAM)-modification for a given peptide would indicate increased irreversible modification.

We detected stimulation-induced, reduction resistant modifications at Cys on more than 200 proteins in WT macrophages. GO analysis of the 149 significantly modified targets (Fig. 4b) revealed proteins involved in metabolic processes, cellular respiration and oxidative pathways (Figs. 4c and S4a–c). Network analysis of protein-protein interactions showed clusters associated with distinct functions, most importantly RNA metabolism, signaling and cellular respiration (Fig. 4d, e). These results indicate that many metabolic proteins are oxidized in inflammatory conditions and these cysteine residues are susceptible to reduction-resistant, and therefore unlikely reversible, modifications. Among these modified proteins, we found

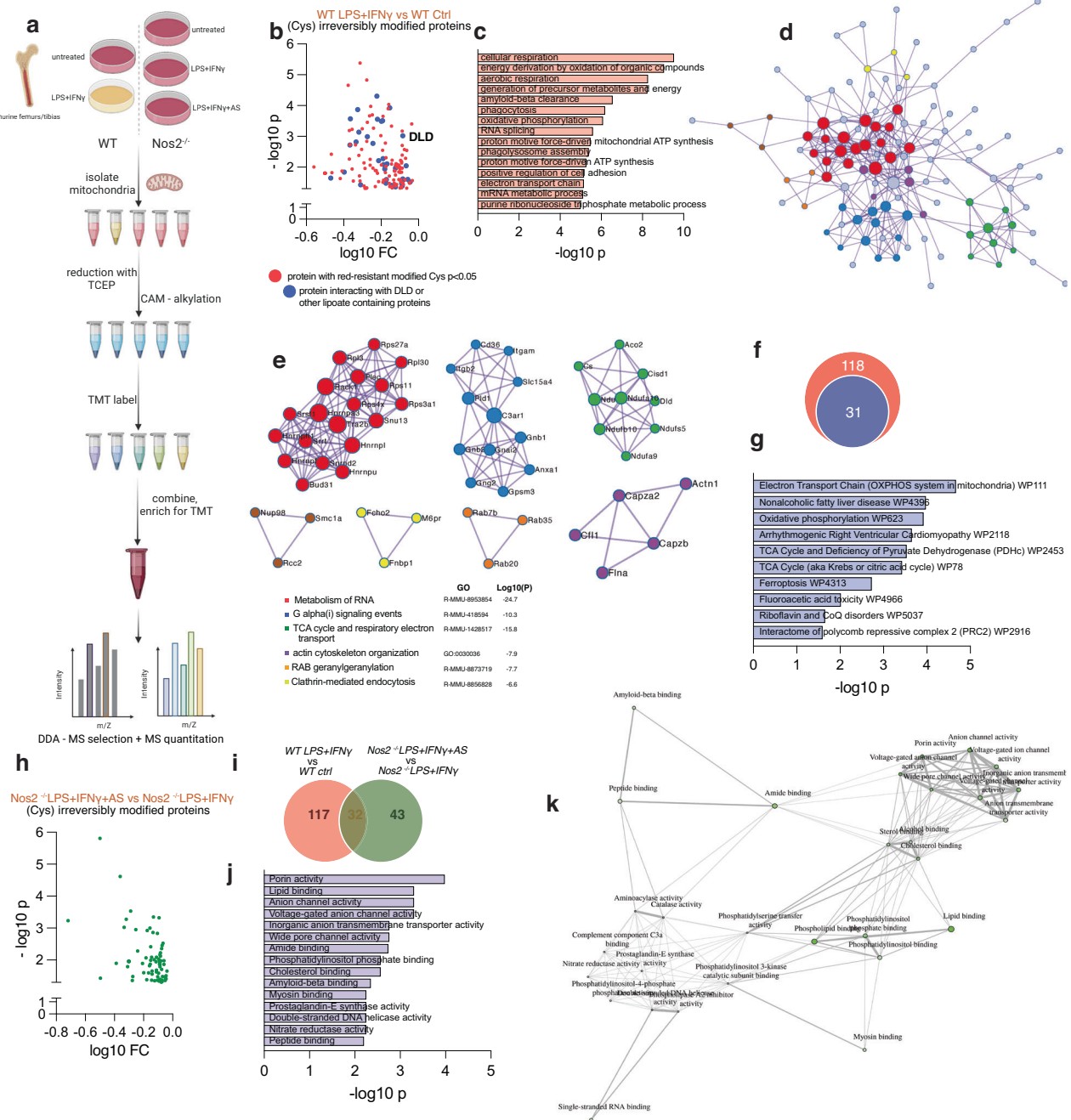

**Fig. 4 | Proteomic analysis reveals stable cys-modifications in M1 BMDMs.**
**a** Experimental design for mitochondrial-enriched proteomics with a focus on non-reversible modifications. **b** Volcano plot showing significant proteins with reduction resistant cysteine modifications with FC < 1 in stimulated WT BMDMs from male and female mice vs ctrl (mixed sex in all groups) (*n* = 4 mice). Data were corrected for total protein abundance. In blue are indicated proteins belonging to interactome of DLD together with DLD itself. **c** Total GO enrichment of proteins from (**b**) (protein number = 149). **d** Proteomics network of protein-protein interactions, depicting proteins as nodes and interactions as edges. **e** Node color is associated with the assigned network cluster. The most enriched GO term associated with each cluster is indicated. **f** Venn diagram indicating number of proteins belonging to interactome of DLD out of total (4b); **g** Enrichment analysis of subgroup of proteins from 4b-f (protein number = 31). **h** Volcano plot showing significant proteins with reduction resistant cysteine modifications with FC < 1 in stimulated *Nos2*−/− BMDMs from male and female mice and treated with AS vs stimulated only (mixed sex in all groups). Data were corrected for total protein abundance (*n* = 4 mice). **i** Venn diagram indicating groups of proteins from (**b**) and (**h**) with respective overlap indicating possible protein modifications driven by endogenous HNO. Pathway enrichment analysis (GO molecular function) (**j**) and network analysis (**k**) of merged list from (**i**). Data shown are representative of two or more independent experiments. Source data are provided as a Source Data file.

DLD with a modification at Cys$^{477}$, together with multiple proteins (31 out of 149, ~20%) belonging to components of the known interactome of DLD (Fig. 4f, g). These are proteins that are likely to be in the proximity of PDH/lipoate-generated HNO. This highlights potential strong susceptibility of DLD, and proteins in its network, to be inhibited by HNO-mediated irreversible oxidation under inflammatory conditions.

We then performed the same proteomic screen using stimulated *Nos2*−/− BMDMs and attributed ~54% of the reduction-resistant proteins in WT BMDMs to NOS2 expression (Fig. S4d). These were

enriched in pathways involving phagocytosis, cellular metabolism and respiration (Fig. S4e, f), with a major subgrouping in organic acid metabolism and, reassuringly, nitrogen-compound metabolic processes (Fig. S4g). Finally, we treated stimulated *Nos2*$^{-/-}$ BMDMs with AS to identify possible direct HNO-modified proteins and found 75 proteins significantly modified (Fig. 4h), 32 of which overlap with reduction-resistant proteins in WT cells (Fig. 4i). Enrichment analysis of these targets showed dominant representation in mitochondrial transport as well as sugar and lipid metabolism related proteins (Fig. 4j, k).

Within DLD, our proteome dataset detected only the CAM-versions of Cys$^{477}$- and Cys$^{484}$-containing peptides indicating reversible modifications. When interrogating NOS2 dependency of modifications, the data on Cys$^{477}$ did not pass our filtering parameters, although CAM-peptides of Cys$^{477}$ in *Nos2*$^{-/-}$ cells indicated decreased modification (Fig. S4h) in the absence of NO.

In addition to identifying targets of modification we used this dataset to gain insight into the biological effects of macrophage exposure to HNO. We assessed relative total abundance of peptide/proteins across the proteome and we measured secreted cytokine/chemokine levels. We found 64 proteins to be significantly upregulated and 47 proteins downregulated in *Nos2*$^{-/-}$ macrophages stimulated in the presence of AS (using a cutoff of FC = 1.25) (Fig. S4i). This target list suggests that the addition of HNO alters the levels of proteins involved mainly in phagocytosis, intracellular transport and autophagy (Fig. S4j, k). Our analysis of secreted factors showed that although cytokines classically associated with macrophage pro-inflammatory activation such as IL-6 and TNFα were not affected by the presence of the HNO, the production of CCL1, IL-1α, C5a, and GCSF was increased with HNO exposure. In contrast, the secretion of CXCL10 and KC was reduced with AS treatment (Fig. S4l). Together these data show that exogenous HNO may tune the inflammatory status of macrophages.

## HNO drives susceptibility of modification at Cys$^{484}$ within DLD

To look more closely at specific effects on DLD, we performed kinetic studies of DLD dehydrogenase activity upon incubation of rhDLD with either HNO or NO donors. These experiments revealed a stronger inhibitory effect of AS (Figs. 5a, b and S5a, b) than DEA/NO, recapitulating what we found in BMDMs (Fig. 2h). This is generally consistent with the reactivity of these related nitrogen oxides. In these assays we also excluded the possible effect of nitrite, a major byproduct of AS decomposition, on DLD activity (Fig. S5a), strengthening our hypothesis that HNO itself is responsible for DLD inhibition.

We next sought to identify post-translational modifications (PTM)s on DLD using high-resolution proteomics and to confirm the presence of HNO-dependent PTM(s). We confirmed that AS-driven modifications would appear as sulfinic acid rather than the Cys sulfinamide; oxidation from sulfinamide to sulfinic acid during proteomics sample prep is indeed a previously described phenomenon[58](Fig. S5c, d). Experiments on recombinant DLD (Fig. S5e) using both label free and TMT, showed five cysteines modified by RNS: Cys$^{80}$, Cys$^{302}$, Cys$^{312}$, Cys$^{477}$ and Cys$^{484}$. Cys$^{80}$, together with Cys$^{302}$ and Cys$^{312}$, were only seen in the sulfonic acid state (+48) (Fig. S5d, f–h) which was increased with all treatments tested when compared to control. These Cys modifications were not specific to HNO, but rather reflect a sensitivity to oxidation. In contrast, Cys$^{477}$ was detected carrying either a sulfinamide (+31), sulfinic acid (+32) or sulfonic acid in the presence of AS or DEA/NO (Figs. 5c and S5i) with sulfinamide and sulfonic acid modifications specific for AS. Assessments of Cys$^{484}$ showed a greater increase in sulfinic acid with AS vs DEA/NO (Fig. 5d). Importantly these two cysteines (Cys$^{477}$ and Cys$^{484}$) were also detected by our TMT assay on BMDMs mitochondrial-enriched proteomes, revealing the importance of these two sites for oxidative modifications in physiological settings.

Here also we excluded nitrite to be responsible for modifying DLD (Fig. S5j).

It is important to note that the pattern of modifications of Cys$^{477}$ and Cys$^{484}$ was different suggesting distinct reactivity. Although both Cys are clearly modified in an AS-dependent manner, Cys$^{477}$ seemed to be particularly sensitive to degrees of oxidation which affected detectability (Fig. S5k). In contrast, Cys$^{484}$ was consistently detected as sulfinic acid-modified across our approaches. Moreover, analysis of the unmodified peptides showed reductions on Cys$^{484}$ in treatments with AS or DEA/NO consistent with reduction-resistant modification (Fig. 5d), while this was not the case for Cys$^{477}$ (Fig. 5c).

As a result of these observations, we focused further analysis of physiologically relevant modifications in BMDMs on Cys$^{484}$ specifically. We used labeled, chemically synthesized Cys$^{484}$ peptides of DLD (Fig. S5a) to track PTM status through our targeted proteomics pipeline. We found that concentrations of AS higher than 0.2 mM result in the highest HNO-specific oxidation state, peaking at modification of ~6% of total protein (Fig. S6b, c). When applying this assay to stimulated BMDMs, we failed to detect Cys$^{484}$ sulfinic acid modified peptides. However, we readily measured reductions in the levels of unmodified peptide when macrophages were stimulated or treated with donors (Fig. 5e) indicating a conversion of these peptides to a reduction-resistant modification state. Importantly, levels of unmodified peptide fall in a NOS2-dependent manner and crucially, *Nos2*$^{-/-}$ cells restored likelihood of modification of Cys$^{484}$ during exposure to HNO (Fig. 5e).

PTMs like lipid peroxidation or modifications with phosphine groups derived from TCEP-RNS interactions were not detected, suggesting they are unlikely to account for the peptide loss of unmodified DLD. While these data do not definitively prove specific HNO-mediated modifications, when cellular GSH was depleted with BSO, pushing the equilibrium of HNO chemistry towards sulfinamide (Fig. 2f), even greater decreases in unmodified Cys$^{484}$ peptide were seen, consistent with increases in modification specifically at this site (Figs. 5f and S6d). We further confirmed this using rhDLD (Fig. 5g) demonstrating that GSH levels control the degree of bioavailability of HNO and likely limit the detectability of modified Cys$^{484}$ in our cellular systems.

Considering this, we contemplated the possibility that the cellular pool of DLD may encompass forms with both reversible and reduction-resistant modifications driven or perturbed by RNS. Therefore, we used IAM-PEG$_2$-biotin tags to alkylate reduced (free) Cys thiols to assess sites that may undergo modification in macrophages, and enriched conjugated peptides were assessed by MS (Fig. S6e). Samples clustered differently at PCA analysis (Fig. 5h, i), highlighting differences induced by stimulation and the presence of NO and HNO. Peptides containing Cys$^{69}$ and Cys$^{484}$ of DLD showed enrichment in this assay. LPS + IFNγ stimulated WT cells had higher levels of IAM-PEG$_2$ tagged Cys$^{484}$ peptides than unstimulated, indicating higher free Cys (Fig. 5j). Interestingly stimulated *Nos2*$^{-/-}$ macrophages showed even higher levels of IAM-PEG$_2$ Cys$^{484}$ peptides, while AS treatment brought these levels down to the stimulated WT. This effect was specific for Cys$^{484}$ (Fig. S6f).

Together these data support the notion that during macrophage stimulation, Cys$^{484}$, and to a lesser extent Cys$^{477}$, of DLD can be modified by HNO resulting in a reduction-resistant PTM detectable as Cys-sulfinic acid. Although this exact modification in intact stimulated cells appears difficult to detect, our data suggest that Cys$^{484}$ may be substantially oxidized in resting macrophages, likely to a disulfide, that may be reduced to modify enzyme activity during stimulation. Moreover, it appears that HNO may dictate the extent to which the disulfide is present. Once available, the increased free Cys$^{484}$ in LPS + IFNγ treated cells is subject to direct modification by HNO resulting in a reduction-resistant drop in PDH activity.

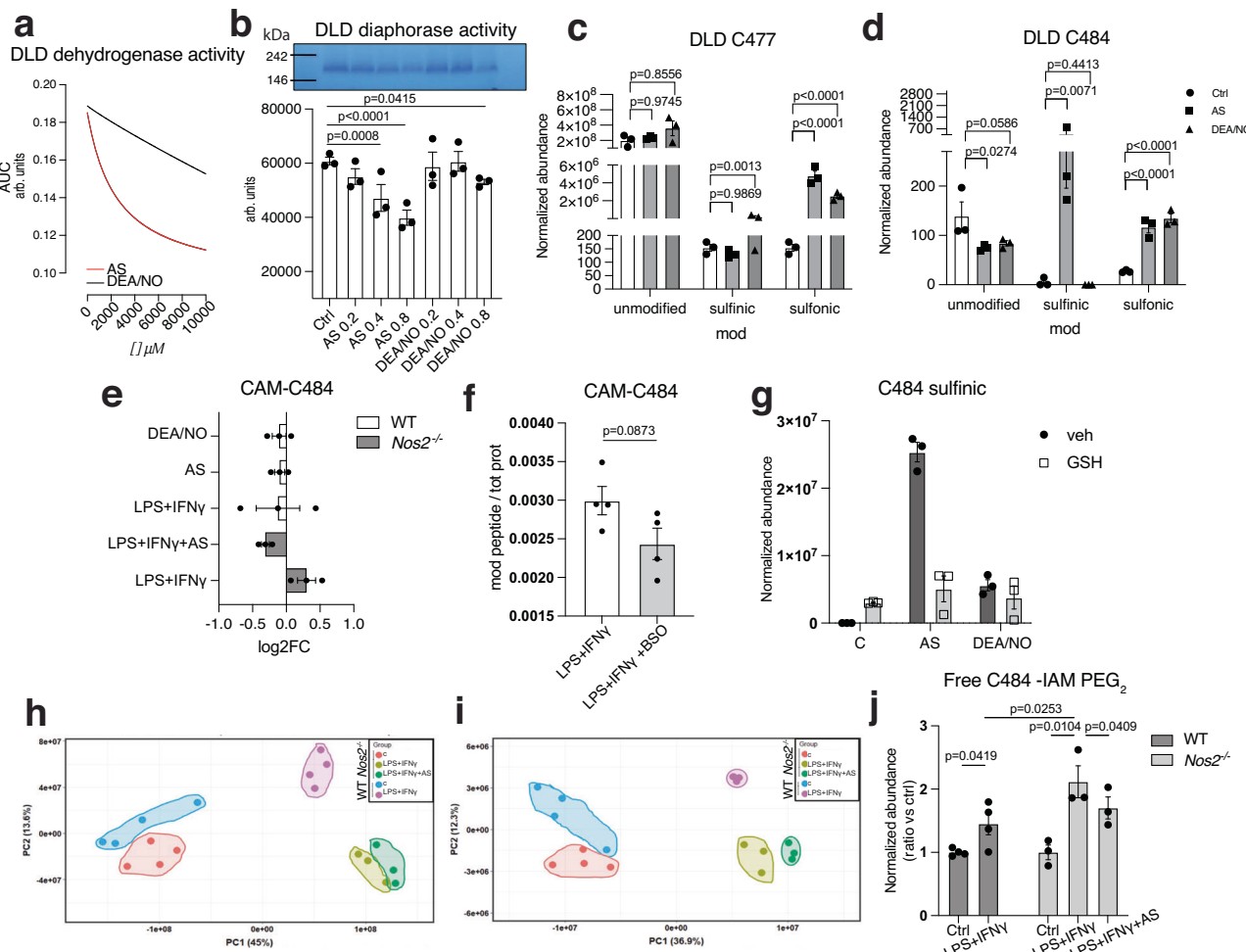

**Fig. 5 | HNO drives susceptibility of modification at Cys[484] within DLD. a** Dose response regression curves for dehydrogenase activity of rhDLD in the presence of AS or DEA/NO ($n = 10$ wells). Data were analyzed by non linear fit (dose-response -inhibition) (R squared = 1 for AS, 0.95 for DEA/NO). **b** Native in-gel activity assay of rhDLD incubated with 500 μM or 2 mM AS or DEA/NO for 1 h at 37 °C. Bar graphs show quantified diaphorase activity ($n = 3$ wells). Proteomics analysis of modifications of rhDLD at Cys[477] (**c**) and Cys[484] (**d**) in the presence of AS or DEA/NO ($n = 3$ wells). Data in (**b, c, d**) were analyzed by one-way ANOVA with Dunnet's multiple comparisons test. **e** Targeted proteomics assay (PRM) on DLD Cys[484]-containing peptide on mitochondrial lysates from WT and *Nos2*[−/−] BMDMs from male and female mice (mixed sex in all groups) stimulated with LPS + IFNγ and AS, indicating Log2FC values (relative to unstimulated cells) of carbamidomethyl (CAM)- Cys[484] containing peptide ($n = 3$ mice). **f** TMT global mitochondrial proteomic analysis on mitochondrial lysates from WT cells from male mice stimulated with LPS + IFNγ for 16 h and treated with BSO for 4 h, indicating normalized abundance of CAM- Cys[484] containing peptide. Data ($n = 4$ mice) were analyzed by unpaired t test (two-tailed). **g** Proteomic LFQ analysis of Cys-sulfinic acid modifications of rhDLD at Cys[484] in the presence of AS or DEA/NO and equimolar GSH ($n = 3$ mice). **h** PCA plots of proteomics data of IAM-PEG₂/TMT labeling of mitochondrial lysates from WT and *Nos2*[−/−] BMDMs from male and female mice (mixed sex in all groups) stimulated with LPS + IFNγ and AS, for identification of free cysteines before (**h**) and after (**i**) peptide enrichment. **j** Quantification of IAM-PEG2-enriched (free) Cys[484]-containing peptide of DLD in conditions from (**h, i**). Basal values in ctrl were set to 1. Data ($n > 3$ mice) were analyzed by two-way ANOVA (Sidak's post-tests). All error bars display mean ± SEM. Data shown are representative of two or more independent experiments. *p* values > 0.05 are reported as "ns" unless otherwise specified. Source data are provided as a Source Data file.

## Cys[477] and Cys[484] sulfinamide impair DLD homodimer formation

To assess how the Cys[484] DLD modifications might affect PDH activity we undertook an in silico modeling approach utilizing available structural data. Current human DLD (hDLD) structures show that known disease-causing mutations map to three locations in the human enzyme: the dimer interface, the active site, and the FAD and NAD (+)-binding sites[59,60]. We used PyMOL to build energetically minimized structural models of the hDLD (PDB_ID: 3rnm) modified at Cys[484] by adding a sulfinamide functional group in place of the native thiol. Our analysis showed that Cys[484] likely forms intra- and inter-chain reciprocal H-bond interactions at the monomer-monomer interface[61] with residues His[364]; Asp[368]; Cys[388]; Pro[390]; Ala[457]; Arg[482]; Val[483]; His[485]. All the above cited residues, alternatively from both chains, locate within 4 Å from Cys[484] from chain A or chain B. Notably, along conformation samplings, Cys[484] and Cys[388] occupy positions less than 2.5 Å distant,

making them likely involved in a possible (transient) interchain disulfide bond.

Our structural analysis shows that introduction of a sulfinamide group at Cys[484] perturbs the local H-bond interaction network and would prevent the formation of a disulfide bridge between Cys[484] and Cys[388]. Specifically, interactions between Cys[484] (chain A) and His[364] (chain B) are weakened (their distance becomes greater than 4 Å) and new interactions with Arg[495] (chain B) are strengthened (Fig. 6a, b). Notably, Cys[484] is the cysteine closest to the FAD cofactor, being only 8 Å away from it (see also Fig. S6g). Cys[477] is the cysteine residue closest to the DLD-E2b subunit (chain E), mapping to 10 Å from it.

We next used the FOLDX repair tool and the Yasara Minimization server to optimize and relax the 3D models, as described in the Methods section. The interaction energies calculated at the monomer-monomer interface within the DLD dimer or at the dimer-monomer

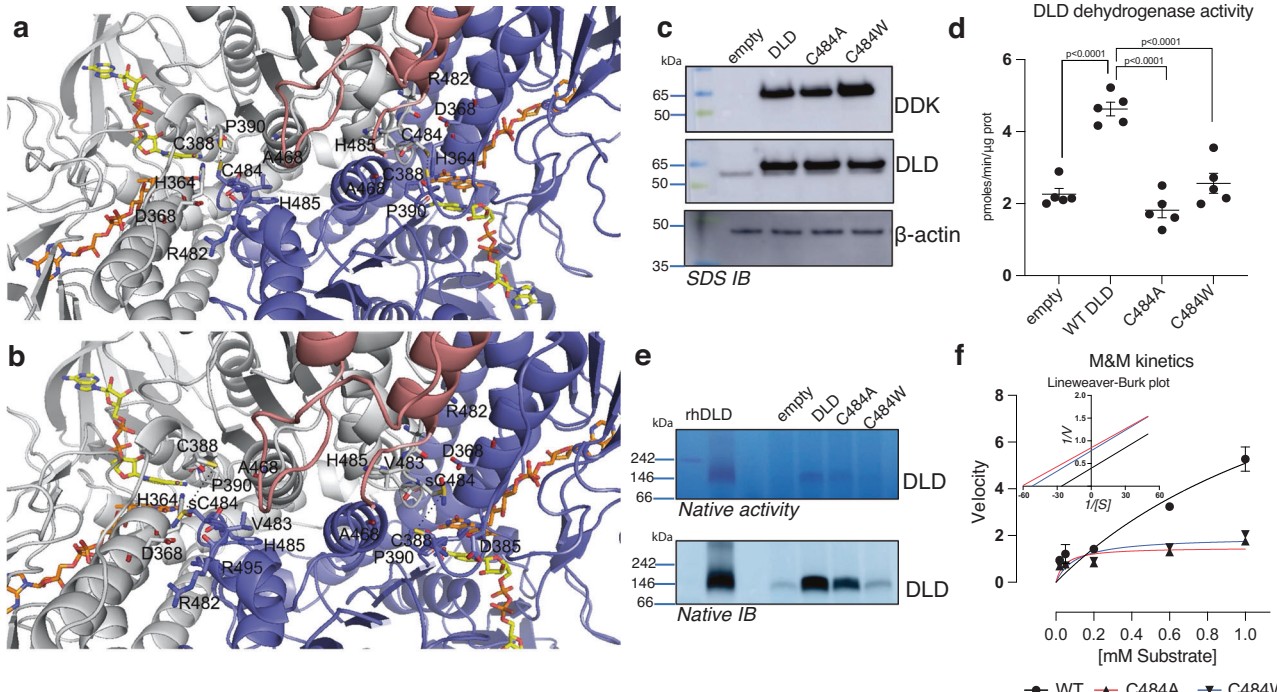

**Fig. 6 | Cys[477] and Cys[484] sulfinamide impair DLD homodimer formation. a** Top view of dihydrolipoamide dehydrogenase (DLD_E3 dimer) in complex with the subunit binding of human dihydrolipoamide transacylase (E2b). DLD_E3 dimer is reported in gray (chain A) and blue (chain B) cartoon representation, whereas E2b is reported in pink cartoon representation. FAD and NAD+ are reported in orange and sticks representation, respectively. Residues within 4 Å from C484 are reported in white (chain A) and blue (chain B) sticks and labeled. Dashed lines indicate the inter-chain interaction between the indicated Cys[388] and Cys[484] residues. The reported distances are lower than 2.5 Å. **b** Residues within 4 Å from Cys[484]sulfinamide (sC484) are reported in white (chain A) and blue (chain B) sticks and labeled. Dashed lines indicate the inter-chain distance at the level of the indicated Cys[388] and sCys[484] residues. The reported distances are greater than 5.5 Å. **c** HEK293T cells were transfected with either empty vector/WT DLD/DLD C484A/DLD C484W. SDS IBs were perfomed probing for the expression tag Myc-DDK, DLD and β-actin as loading control. **d** Whole cells lysates where assessed for DLD dehydrogenase activity (*n* = 5 technical repeats). Data were analyzed by one-way ANOVA with Dunnet's multiple comparisons test. **e** DLD in-gel activity assay and IBs on native gels of whole cells lysates from HEK293T transfected as in (**c**). **f** Lysates from (**d**) were used for kinetic assay for DLD dehydrogenase activity with increasing concentration of lipoic acid (*n* = 4 technical repeats). Non linear regression model of Michealis-Menten is shown, together with Lineweaver-Burk plot. All error bars display mean ± SEM. Data shown are representative of two or more independent experiments. Source data are provided as a Source Data file.

interface within the DLD E3-E2b trimer, both in the unmodified and in the DLD E3-E2b Cys[484] sulfinamide modified complex, gave a negative binding energy value (Table 1) that was lower in the native configuration than in the modified complex. This confirmed the expectation of a weakened binding interaction in presence of the modified Cys[484] sulfinamide residue. This was further confirmed also by relaxing the models in the Rosetta force field (Table 2).

Lastly, we deployed the "residue_energy_breakdown" tool and confirmed reciprocal monomer-monomer interchain interactions between Cys[484] and His[485], Ala[486] and Arg[495] residues (Supp. Table 1) are weakened in the presence of the sulfinamide group. Notably, the introduction of sulfinamide modification at Cys[484] appears to weaken interactions between Cys[484] of chain A and residues Asp[368], Val[389] and Pro[390] of chain B (Supp. Table 1) even more substantially. The presence of a sulfinamide on Cys[447] slightly decreases the interaction energy at the DLD monomer-monomer interface and slightly increases it at the interface of DLD E2b subunit with the DLD dimer (Supp. Table 2), perhaps due to a different rearrangement of the local H-bond interaction network. This different rearrangement causes a different orientation of several side-chains, including the side-chains of Cys[484] and Cys[388] (Fig. S6g).

A role for Cys[484] in DLD activity is totally unexplored experimentally. Therefore, we changed Cys[484] to alanine (C484A) to eliminate the possibility of disulfide linking without steric separation of DLD dimers, or tryptophan (C484W) to eliminate disulfide linking and exert a steric force to interfere with non-covalent dimerization. In silico modeled

interaction energies supported these selections (Supp. Table S3). Transient overexpression of Myc-DDK tagged proteins in HEK293T cells showed that the mutations did not prevent the expression of the DLD protein (Fig. 6c). In addition, dehydrogenase activity was almost completely ablated in both mutants (Fig. 6d) proving the fundamental role of this residue in DLD function. Moreover, in native gels, the mutant proteins showed reduced ability to form dimers, and in gel diaphorase activity was largely absent (Fig. 6e). Interestingly, while dehydrogenase activity was comparably reduced in both mutants, the C484A mutant partially retained some dimerization capacity as opposed to C484W. Both mutations lead to lower Vmax values, indicating reduced catalytic efficiency, as represented by the classical Michaelis-Menten curve (Fig. 6f). Further enzyme kinetics showed that these mutants behave as if the enzyme is in the presence of an uncompetitive inhibitor, as shown by the Lineweaver–Burk plot, consistent with an enzyme-substrate complex that is highly stable, preventing the release of the product and slowing down the reaction (Fig. 6f).

## Hepatocyte PDH is regulated in NO-dependent manner consistent with the generation of HNO

Utilization of lipoylated PDH and the generation of NO are not characteristics limited to macrophages. Therefore, we investigated the possibility that PDH-generated HNO may modify enzyme function in other environments. Hepatocytes upregulate NOS2 and produce NO under certain conditions[62–64]. Thus, as a first approach we used liver

**Table 1 | Energy calculations about the investigated 3D models**

| Interaction energies (FoldX AnalyseComplex) | WT | | C484sulfinamide modified DLD | |
|---|---|---|---|---|
| Evaluated parameters | InterChain.A_B | InterChain.AB_E | InterChain.A_B | InterChain.AB_E |
| Group1 | A | AB | A | AB |
| Group2 | B | E | B | E |
| IntraclashesGroup1 | 29,5201 | 68,3825 | 30,0418 | 70,7886 |
| IntraclashesGroup2 | 34,9679 | 2,75887 | 36,619 | 2,90248 |
| **Interaction Energy** | **−90,6667** | **−9,56689** | **−88,5912** | **−9,45471** |
| Backbone Hbond | −20,4434 | −1,89106 | −18,8878 | −1,892 |
| Sidechain Hbond | −40,3367 | −6,84229 | −37,4774 | −6,75942 |
| Van der Waals | −73,7061 | −7,66983 | −73,7899 | −7,63457 |
| Electrostatics | −8,97054 | −2,11671 | −9,12802 | −2,12716 |
| Solvation Polar | 93,908 | 14,3846 | 93,3047 | 14,3999 |
| Solvation Hydrophobic | −100,265 | −9,20305 | −100,58 | −9,10423 |
| Van der Waals clashes | 2,42316 | 0,207425 | 2,54317 | 0,201281 |
| entropy sidechain | 42,492 | 4,53423 | 40,836 | 4,41639 |
| entropy mainchain | 15,6268 | 1,0743 | 15,9141 | 1,07426 |
| torsional clash | 1,47145 | 0,221081 | 1,58459 | 0,205195 |
| backbone clash | 14,7392 | 1,9886 | 14,7351 | 1,98845 |
| helix dipole | −1,81962 | −1,55306 | −1,6856 | −1,53745 |
| electrostatic kon | −1,76226 | −0,712563 | −1,92266 | −0,696887 |
| energy Ionization | 0,715411 | 7,77E−16 | 0,698004 | −5,55E−16 |
| Entropy Complex | 2384 | 2384 | 2384 | 2384 |
| Number of Residues | 994 | 994 | 994 | 994 |
| Interface Residues | 195 | 29 | 197 | 29 |

Chain A and B indicate DLD_E3 chain within 3rnm.pdb and in the corresponding Cys[484]-sulfinamide modified protein. Chain E indicates the E2b subunit within 3rnm.pdb and in the corresponding Cys[484]-sulfinamide modified protein. Bold numbers indicate interdomain interaction energies of energetically relaxed protein complexes.
"PDB.Chain" indicates the chain of the PDB used within the indicated analyses on the cited crystallized structures or models obtained as described in the Methods section.

mitochondria to assess the metabolic response to NO-related donor compounds in vitro. We found that both AS and DEA/NO were capable of decreasing liver PDH activity (Fig. 7a, b) with AS being a more powerful inhibitor, consistent with observed effects in BMDMs and rhDLD (Fig. 2h and Fig. 5a, b). Moreover, electron flow assays showed that hepatocyte pyruvate respiration is compromised strongly with AS (Fig. 7c) whereas the activity of other ETC complexes was not affected (Fig. 7d, e).

Based on our identification of DLD reacting with HNO, we tested CI respiration from fuel sources dependent on OGDH[65,66]. Here, too, the effect of HNO was detectable (Fig. S7a, b). To address whether HNO-mediated inhibition of PDH occurred in vivo, we assessed hepatic PDH status in endotoxin-treated mice (Fig. 7f) and found a decrease in PDH activity in liver mitochondria of these mice (Fig. 7g, h). Moreover, similar to macrophages, we found these reductions to be resistant to reversal with DTT, Cys or GSH (Fig. 7i). Studies in Nos2[−/−] mice confirmed this effect on PDH to be NOS2 dependent (Figs. 7j–l and S7c).

To dissect whether hepatocyte intrinsic Nos2 expression is capable of inhibiting PDH we isolated primary cells from perfused murine livers and confirmed their ability to produce NO at comparable levels to those in macrophages (Fig. 7m). Here, also we found that stimulated cells have compromised pyruvate-elicited OCR (Figs. 7n and S7d), and decreased PDH activity (Figs. 7o and S7e). Cytokine-induced NOS2 was even more effective than lower doses of NO donors (Fig. 7p). In parallel, we extracted metabolites from these primary cultures and evaluated GSH-sulfinamide using ESI-LC/MS-MS. We found that cells treated with cytokines accumulated GSH-sulfinamide as did cells treated with HNO or NO donors and pre-treatment with AG prevented this phenomenon (Figs. 7q and S7f). Lastly we confirmed in this system that, as predicted, high intracellular GSH[67] limits HNO-derived sulfinamide (Fig. S7g–j).

Taken together these data show that modifications of PDH activity consistent with HNO chemistry are generated in hepatocytes during sepsis as a result of endogenous, NOS2-dependent NO and that intracellular levels of GSH are critical in influencing the development of HNO-derived reduction-resistant modifications. This suggests that the PDH-derived HNO generation we propose here is applicable across various cell types as long as the GSH environment is amenable.

## Discussion

Multiple groups have shown the PDH and OGDH complexes to be targeted during proinflammatory stimulation of macrophages, and some have attributed this finding to the loss of the lipoate cofactor without a well-defined mechanism or consideration of involvement of RNS[47]. PTMs via the kinase PDK have also been suggested[12,68]. Nevertheless, we and others have shown that although relevant, this phenomenon does not account for the metabolic reprogramming at PDH in macrophages[6,47,69]. The mechanism we describe here is, to our knowledge, previously unrecognized and reveals a biochemical link between reduced radicals of NO and cellular metabolism, where a key role is played by the microenvironment. Our data suggest a model where, in the presence of NO, the thiol-rich environment represented by the multiple lipoate moieties of PDH-E2, acts as an intramolecular HNO generator (Fig. 8). Within the PDH complex, this HNO is produced in close proximity to the cysteines of DLD resulting in rapid reaction of HNO with the DLD thiols leading to stimulation-dependent, reduction-resistant, sulfinamide modification and inhibition. Molecular modeling and mutagenesis confirm that modification at Cys[484] interferes with the dimeric composition of DLD while shutting down enzymatic activity. This demonstrates physiologically relevant generation of HNO, highlighting a potential role for this species in metabolism and biology.

**Table 2 | The PDB.Chains used in the analyses with Rosetta were the same indicated in Table 1**

| INTERFACEANALYZER APP | WT.A_B | WT.AB_E | C484.sulfinamide.A_B | C484.sulfinamide.AB_E |
|---|---|---|---|---|
| complex_normalized | 0829 | 0921 | 0878 | 0932 |
| **dG_separated** | **−164,508** | **−14.133** | **−157.316** | **−14.048** |
| dG_separated/dSASAx100 | −2066 | −1.237 | −1.975 | −1.235 |
| dSASA_hphobic | 4412,5 | 631.604 | 4.469.150 | 621.870 |
| dSASA_int | 7962,872 | 1.142.113 | 7.967.294 | 1.137.917 |
| dSASA_polar | 3550,372 | 510.508 | 3.498.144 | 516.047 |
| delta_unsatHbonds | 34 | 3.000 | 33.000 | 1.000 |
| dslf_fa13 | 1139 | 1.139 | 1.139 | 1.139 |
| fa_atr | −6083,011 | −6.081.043 | −6.096.449 | −6.096.449 |
| fa_dun | 2707,8 | 2.707.800 | 2.712.322 | 2.712.322 |
| fa_elec | −1566,988 | −1.567.587 | −1.562.165 | −1.562.165 |
| fa_intra_rep | 12,356 | 12.357 | 12.705 | 12.705 |
| fa_intra_sol_xover4 | 215,991 | 215.991 | 216.419 | 216.419 |
| fa_rep | 5734,169 | 5.732.690 | 7.159.078 | 7.159.078 |
| fa_sol | 3821,65 | 3.816.065 | 3.828.241 | 3.828.241 |
| hbond_E_fraction | 0242 | 0204 | 0257 | 0205 |
| hbond_bb_sc | −150,436 | −150.436 | −150.722 | −150.722 |
| hbond_lr_bb | −238,149 | −238.149 | −238.149 | −238.149 |
| hbond_sc | −78,691 | −78.691 | −77.714 | −77.714 |
| hbond_sr_bb | −287,354 | −287.354 | −287.354 | −287.354 |
| hbonds_int | 43 | 4.000 | 42.000 | 4.000 |
| lk_ball_wtd | −61,909 | −61.907 | −63.055 | −63.055 |
| nres_all | 940 | 991.000 | 940.000 | 991.000 |
| nres_int | 274 | 51.000 | 274.000 | 51.000 |
| omega | 137,884 | 167.863 | 167.863 | 167.863 |
| p_aa_pp | −168,903 | −167.453 | −167.453 | −167.453 |
| packstat | 0636 | 0513 | 0618 | 0618 |
| per_residue_energy_int | 0,39 | 0826 | 0424 | 0857 |
| pro_close | 684,611 | 684.611 | 684.611 | 684.611 |
| rama_prepro | 159,539 | 170.234 | 170.234 | 170.234 |
| ref | 453,216 | 453.216 | 453.216 | 453.216 |
| sc_value | 0666 | 0491 | 0656 | 0489 |
| side1_normalized | 0249 | −0212 | 0322 | −0159 |
| side1_score | 34,109 | −6.157 | 44.114 | −4.615 |
| side2_normalized | 0531 | 2.195 | 0526 | 2.197 |
| side2_score | 72,793 | 48.285 | 72.095 | 48.342 |

A list of the energy terms taken from https://www.rosettacommons.org/docs/latest/application_documentation/analysis/interface-analyzer follows. Energy term abbreviation meanings: dslf_fa13 indicates disulfide geometry potential; fa_atr indicates Lennard-Jones attractive between atoms in different residues; fa_dun indicates the internal energy of sidechain rotamers; fa_elec indicates coulombic electrostatic potential with a distance-dependent dielectric; fa_intra_rep indicates Lennard-Jones repulsive between atoms in the same residue; fa_rep indicates Lennard-Jones repulsive between atoms in different residues; fa_sol indicates Lazaridis-Karplus solvation energy; hbond_bb_sc indicates sidechain-backbone hydrogen bond energy; hbond_lr_bb indicates backbone–backbone hbonds distant in primary sequence; hbond_sc indicates sidechain–sidechain hydrogen bond energy; hbond_sr_bb indicates backbone–backbone hbonds close in primary sequence; pro_close indicates Proline ring closure energy and energy of psi angle of preceding residue; rama indicates Ramachandran preferences; ref indicates reference energy for each amino acid; complex_normalized indicates the average energy of a residue in the entire complex; dG_separated, reported in bold characters, indicates the change in Rosetta energy when the interface forming chains are separated (binding energy), versus when they are complexed. dSASA_int, indicates the solvent accessible area buried at the interface, in square Angstroms. dG_separated/dSASAx100, separated binding energy per unit interface area × 100 to make units fit in score file. Scaling by dSASA controls for large interfaces having more energy; delta_unsatHbonds indicates the number of buried, unsatisfied hydrogen bonds at the interface; hbond_E_fraction indicates the amount of interface energy (dG_separated) accounted for by cross interface H-bonds; hbonds_int indicates the total cross-interface hydrogen bonds found; nres_all indicates the total number of residues in the entire complex; nres_int indicates the number of residues at the interface; per_residue_energy_int; indicates the average energy of each residue at the interface; side1_score indicates the energy of one side of the interface; side2_score indicates the energy of the other side of the interface; side1_normalized indicates the average per-residue energy on one side of the interface; side2_normalized indicates the average per-residue energy on the other side of the interface. Rosetta energy terms are expressed in Rosetta Energy Units (REU) according to https://www.rosettacommons.org/docs/latest/rosetta_basics/Units-in-Rosetta.

Our model of generation of HNO by the lipoate moieties of PDH is supported by multiple lines of evidence. (1) We extend previous work showing disruption of PDH in activated macrophages[6,47] and of modifications of lipoate levels[46], and we can reconsitute pyruvate oxidation with exogenous lipoate. (2) Proteomics of lipoate status in cells exposed to NO is uniquely consistent with NO-lipoate chemistry. (3) We detected HNO-mediated modifications on trapping GSH when NO donors reacted with reduced lipoate. (4) Proteomics show reduction-resistant Cys modifications in recombinant protein and activated macrophages consistent with HNO. (5) Use of HNO detectors confirmed lipoate-dependent production of HNO in macrophages. Thus, we propose that the PDH complex might represent a particularly efficient HNO generator.

In the past decade, research on HNO has revealed a distinct bioactivity profile compared to NO. The primary cellular targets for HNO are thiols and oxidized metals while NO principally interacts with

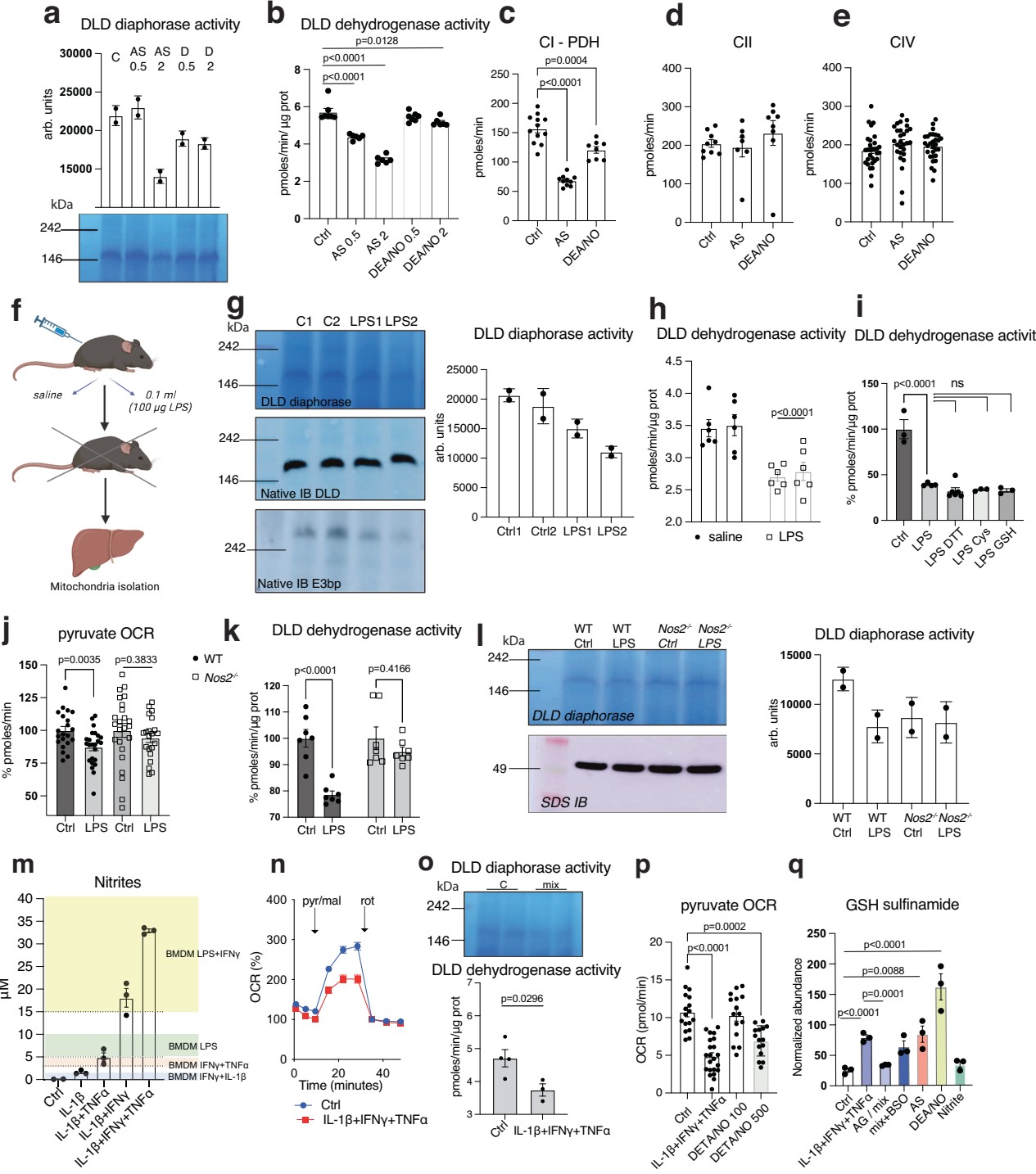

**Fig. 7 | Hepatocyte PDH is regulated in NO- and HNO- dependent manner.** Liver mitochondria from WT male mice were incubated with AS or DEA/NO; bar graphs show DLD diaphorase (**a**) (*n* = 2 technical repeats), dehydrogenase (**b**) activity (*n* = 6 mice) and Complex I, CII, and CIV activities (*n* > 8 wells) (**c**–**e**). Data in (**b**–**e**) were analyzed by one-way ANOVA with Dunnet's multiple comparisons test. **f** Male and female mice (mixed sex in all groups) were injected i.p. with LPS for 24 h. **g** DLD in-gel activity and DLD / E3bp native IB, **g**, **h** quantified DLD diaphorase (*n* = 2 technical repeats) and dehydrogenase activity (*n* = 6 mice per genotype) (two-way ANOVA, Sidak's post-tests) from mice in (**f**). **i** Liver mitochondria lysates from (**f**–**h**) were incubated with reducing agents before assessment of DLD dehydrogenase activity (*n* > 3 mice) (one-way ANOVA with Dunnet's multiple comparisons test). **j**, **k** Quantified pyruvate respiration (*n* > 21 wells) as in (**c**) and DLD dehydrogenase activity in liver mitochondria from WT and *Nos2*⁻/⁻ injected with LPS (*n* = 7 mice per genotype) (two-way ANOVA, Sidak's post-tests). **l** DLD in-gel activity and native IB in

liver mitochondria from (**j**, **k**) (*n* = 2 technical repeats). **m** Nitrite levels in supernatants of primary hepatocyte cultures from male mice (*n* = 3 mice). Color shading represent range of nitrite production in media of BMDMs stimulated as indicated for comparison. **n** Seahorse analysis of pyruvate respiration in primary hepatocytes (as in 1b, c) stimulated with cytokine combination (*n* > 18 wells). **o** DLD in-gel and dehydrogenase activity on mitochondrial extracts from primary hepatocytes (*n* > 3 mice). Data were analyzed by unpaired t-test (two-tailed). **p** Quantified pyruvate respiration in primary hepatocytes stimulated with cytokine combination or DETA/NO (*n* > 15 wells) (one-way ANOVA with Dunnet's multiple comparisons test). **q** GSH-sulfinamide was quantified by ESI-LC/MS-MS from primary hepatocytes cultures (*n* = 3 mice) (one-way ANOVA with Dunnet's multiple comparisons test). All error bars display mean ± SEM. Data shown are representative of two or more independent experiments. *p* values > 0.05 are reported as "ns" unless otherwise specified. Source data are provided as a Source Data file.

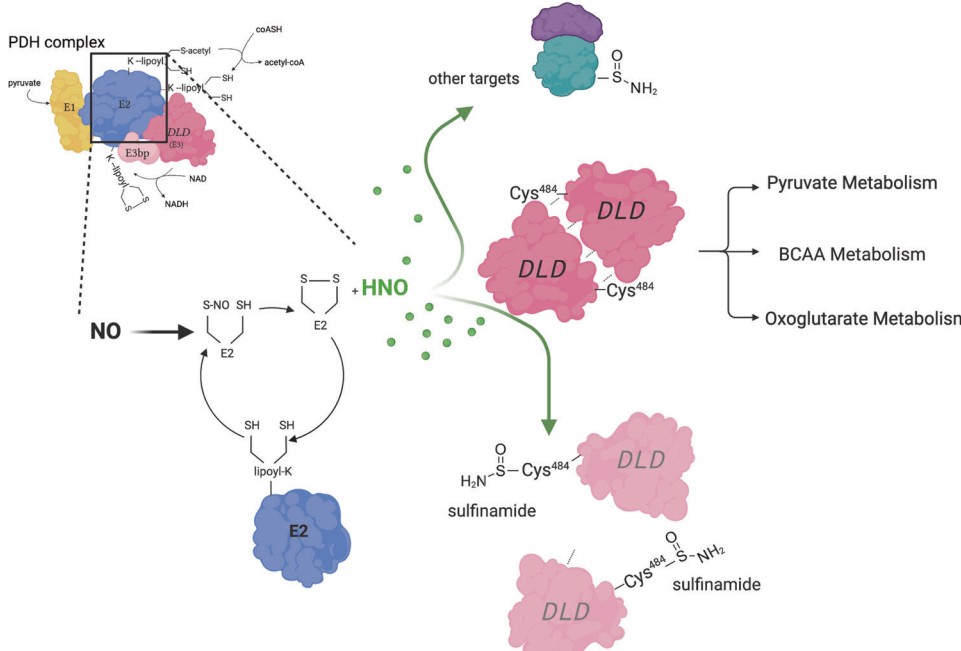

**Fig. 8 | Schematic model for in situ HNO generation.** NO production of M1 macrophages together with the substantial number of lipoate groups in the PDH complex are conditions ripe for HNO generation and signaling. The interaction of NO with reduced lipoate-bound to PDH-E2 is likely to generate HNO together with oxidized lipoate as a resulting product. HNO resulting from exposure of NO to PDHE2-, will promptly target DLD due to proximity, together with other cellular proteins. Reaction with DLD thiols will form a sulfinamide modification. Modified Cys[484] impairs dimerization and DLD activity compromising the entire PDH complex function. Since other mitochondrial dehydrogenases share DLD, this mechanism leads to repercussions on other major intracellular metabolic pathways. PDH pyruvate dehydrogenase, DLD dihydrolipoamide dehydrogenase, BCAA branched chain amino acid.

other free radicals and reduced metals[29,70]. Unlike NO, the high reactivity of HNO for thiols will limit diffusion such that HNO will likely need to be produced in close proximity to its target. Compartmentalization of HNO targets within organelles like mitochondria has been proposed as a way to promote reactivity by avoiding scavengers such as GSH[20]. Still, unequivocal evidence for physiological generation of HNO and its mechanism of action remained elusive until now.

According to models of PDH structure, inter-lipoyl-domain coupling might help in shuttling the reducing equivalents from E1 to E3[71–73] representing a supportive context for endogenous HNO production. The patterns of E3bp modification observed in native gels support current models of the complex[53], and we find that RNS perturb clusters of E3bp:E3 dimers, possibly involving modifications of Cys on the E3 dimer and/or E3bp- bound lipoate.

Using a broad approach, we identify a spectrum of proteins that are modified in a reduction-resistant manner during macrophage stimulation and ascribe patterns of irreversibility to NOS2. These findings are in line with a previous ex vivo study[58], and our work moves this forward by demonstrating production of HNO in cells and describing a likely model for its involvement in metabolic rewiring. Although we could not identify the exact modifications on DLD in cellular settings, we conclude that NOS2-mediated signaling during inflammatory activation may drive oxidation of these targets via reduced RNS, as the pattern of CAM-modified peptides in BMDMs suggests.

Our experiments with recombinant DLD identified Cys[484] as a likely target of reaction with lipoate-generated HNO in PDH. Computational analysis of DLD revealed that introduction of a sulfinamide group at Cys[484] would be expected to perturb interactions both at the interface of the DLD monomers and at the interactions between the DLD dimer and human dihydrolipoamide transacylase (E2b). We also found that Cys[484] may form a disulfide inter-chain with Cys[388] that would be disrupted by sulfinamide on Cys[484]. This possibility is supported by our findings from our IAM-PEG$_2$ dataset on free cysteines

and by our observations of CAM-Cys[388] peptide and its apparent similarity to other disulfides (Fig. S6h–i). Notably, the predicted disulfide is only 12 Å from FAD, suggesting that it might be involved in the electron tunneling phenomena[74,75] involved in the reoxidation of FADH$_2$[76,77]. We then confirmed the importance of Cys[484] using mutagenesis by showing that alanine or tryptophan substitutions ablated DLD activity and altered dimerization capability, therefore demonstrating Cys[484] to be crucial for the proper functioning of DLD.

Our implication of Cys[477] and Cys[484] of DLD in HNO-mediated inhibition of PDH is supported by proteomic studies that revealed them as susceptible to modification[78], but our data demonstrate importance of these cysteines as sites of physiologically relevant, reduction-resistant modification in proinflammatory macrophages, and define their involvement in regulation of enzyme activity.

In total, our data show that it is very likely that local HNO production takes place in environments around lipoate and may involve not only DLD but other proteins associated with DLD or belonging to its interactome. This would drive irreversible modifications and alterations of activity. Moreover, the finding of a lower $K_i$ for HNO than NO on PDH underlines the importance of the localization in driving inhibition. Location of HNO production together with target protein environments have been suggested as potential mechanisms controlling HNO specificity. While GSH concentrations can affect HNO-mediated inhibition[55,79,80], protein structures that result in lower thiol pK$_a$ values may facilitate reaction with HNO even in the presence of relatively high levels of GSH.

It should be noted that the mechanism we describe here would suggest that HNO reacts with other mitochondrial dehydrogenases sharing the DLD subunit (Fig. S2a). Other groups have found OGDH activity to be inhibited in a NO-dependent manner in macrophages, and we report here its sensitivity to HNO. In addition, we use isotopic labeling experiments to demonstrate that BCKDH is also inhibited in a NOS2-dependent way. However, no one has ascribed this action to

HNO generation before or reported the inhibition of these enzymes to be due to DLD subunit inactivation specifically. The kinetics of inhibition of PDH vs OGDH, BCKDH and others may be different considering cellular localization but we propose that the key consideration is the proximity of lipoyl moieties to facilitate the generation of, and then reaction with HNO[81,82].

In summary, our data provide compelling evidence of a mechanism of intramolecular HNO production by the lipoate moieties associated with PDH. We document a physiological generation of HNO that does not involve $H_2S$[83], and suggest that HNO is critical in the inhibition of PDH flux mediated by NO working in cis- and trans- in primary macrophages and hepatocytes. Although treatment with AS revealed additional putative protein targets of endogenously produced HNO (Fig. 4j, k), direct parallels are difficult to make given the relative lack of information regarding physiological HNO levels and/or location. Regardless, our data suggest that HNO may not be strictly involved in the classical inflammatory capacity of macrophages per se, but rather this species may have distinct roles in regulating aspects of intracellular transport, ER homeostasis and cell migration, which are important in tuning inflammatory response and/or directing response towards resolution. Further studies will be required to examine the full implications of this in situ HNO production and to address the proportion of protein targets that are dependent specifically on lipoylated-protein -mediated HNO production.

## Methods

### Mice
Wild-type C57BL/6J mice (strain #:000664) and $Nos2^{-/-}$ (strain #002609) were obtained from The Jackson Laboratory and bred and maintained at the NCI Frederick Cancer Research and Development Center in specific pathogen-free/SPF conditions at ambient temperature and humidity. Animal care was provided in accordance with the procedures in, *"A Guide for the Care and Use of Laboratory Animals"*. Ethical approval for the animal experiments detailed in this manuscript was received from the Institutional Animal Care and Use Committee (Permit Number: 000386) at the NCI-Frederick. All mice used were used between 6 and 10 weeks old and were age- and sex-matched for each experiment. All mice were euthanized using carbon dioxide and cervical dislocation.

### Macrophage culture and stimulation
For murine BMDM isolation, total cells from the bone marrow were plated at $7.0 \times 10^5$/mL for 5–7 days in complete Dulbecco's modified Eagle medium (DMEM) (Gibco #11965) supplemented with 2 mM Glutamine, 10% FBS, 100 U/mL penicillin/streptomycin and 20 ng/ml mouse M-CSF allowing for BMDM differentiation and growth[3]. After a 6-day incubation, BMDM were placed in complete culture media and treated with vehicle (PBS) or 100 ng/mL of LPS E. coli Olll:B4 (Sigma Aldrich) or a combination of LPS (100 ng/mL) and 50 ng/mL of IFNγ for 4–18 h. "Activation", "stimulation" and "(M1) polarization" in the manuscript are referred to LPS + IFNγ treatment, unless specified. Where indicated, cells where pretreated with Aminoguanidine (AG; 1 mM, Sigma Aldrich) or ML226 (3 μM, Cayman Chemical) for 1 h before stimulation. In culture treatments with Lipoic acid (200μM, Sigma T1395) or Buthionine-sulfoximine (BSO; 500 μM Sigma Aldrich) were performed respectively for 2 h and 4 h before harvest. Resting cells where additionally treated with DETA/NO (NO donor)(100 up to 500 μM) for 18 h, DEA/NO (NO donor) or Angeli's Salt (AS) (HNO donor) (400μM or otherwise specified, chemically synthesized in house and prepared fresh in 10 mM NaOH) (4–18 h) where indicated.

### Cell lines
HEK293T cells were purchased from ATCC. Cells were propagated in DMEM with 4.5 g/L glucose, supplemented with 10% fetal bovine serum (FBS) and 2 mM glutamine, at 37 °C and 5% $CO_2$ in a humidified incubator.

### Isolation of hepatocytes and culture
Mouse hepatocytes were isolated from male mice as previously described using a modification of the in situ collagenase (type I, Worthington) perfusion technique of Seglen[62,84].

Hepatocytes ($0.25 \times 10^6$/ml) were plated onto Collagen I-coated dishes and maintained in DMEM with L-arginine (0.50 mM), insulin-transferrin-selenium (Thermo Fisher), L-glutamine, penicillin, streptomycin, and 5% FBS. After a 3-h incubation at 37 °C, the media was changed and the following agents added at the indicated dose and duration: murine recombinant TNFα (R&D) (20 ng/mL), murine recombinant IFNγ (Peprotech) (50 ng/mL), IL1β(R&D) (20 ng/mL), LPS (1μg/mL), or DETA/NO (100–500 μM). "Mix" in the figures refers to IL-1β + IFNγ + TNFα combination treatment. Treatments with AG, BSO, DEA/NO, and AS were performed as in BMDMs. After stimulation for 24 to 48 h, hepatocytes were harvested in Trypsin 0.05%, cell supernatants were assayed for nitrites and cell pellets processed for metabolic analyses.

### Sepsis in mice
Male and female mice (mixed sex in all groups) were challenged i.p with 100 μg LPS (0.1 ml of a solution of LPS in saline) (4 mg/kg) or saline control injection. The liver was harvested after 24 h.

### NO detection
Nitrite, the oxidation product of NO, was measured in cell supernatants using a Griess reagent kit according to the manufacturer's instructions (Invitrogen™, #G7921).

### Immunoblotting
Protein expression was determined by western blot as previously described[3] using 20 μg of total protein per sample. SDS immunoblots were performed to evaluate steady-state protein levels. Antibodies in this study are as follows: Anti-Pyruvate Dehydrogenase E2 antibody (1:1000, Abcam, #ab172617) Anti-Lipoamide Dehydrogenase antibody (1:3000, Abcam, #ab133551), Anti-Lipoic Acid antibody (1:3000, EMD Millipore #437695), Anti -PDHX (1:1000, R&D systems AF6014), Anti DDK (Myc-DKK Tag) (1:5000, Origene #TA50011). For loading control Anti TOM20 (1:5000, Santa Cruz, #sc-136211) and Anti β-actin (1:3000 Abcam, #ab6276) were used.

### Spectrophotometric measurement of total pyruvate dehydrogenase
For mitochondrial enzymatic activity measurements, $20 \times 10^6$ cells were permeabilized beforehand with 0.007% digitonin as described[6] to remove cytoplasmic proteins. Formation of NADH was used to measure pyruvate dehydrogenase complex activity[85]. The reaction mixture here contained 2.5 mM NAD, 0.1 mM Coenzyme A, 5 mM pyruvate and 1 mM $MgCl_2$. A baseline was determined with sample and reference both containing the entire reaction mixture except for pyruvate. Spectrophotometric readings were measured every 20 s for 30 min.

### Metabolic analyses
OCR and ECAR were examined using the XF96 Seahorse Metabolic Analyzer from Seahorse Biosciences (North Billerica, MA)[65]. Briefly, BMDM at day 7 of differentiation and activated for 16 h or primary hepatocytes at day 2 of culture were plated at a seeding density respectively of $0.8 \times 10^5$ cells/well and $0.15 \times 10^5$ cells/well in 200 μl of complete medium. Upon XF96 run, media was replaced with MAS buffer (220 mM Mannitol, 70 mM Sucrose, 10 mM $KH_2PO_4$, 5 mM $MgCl_2$, 2 mM HEPES, 1 mM EGTA, 0.2% Fatty Acid Free BSA). Oxygen

consumption was monitored permeabilizing cells with 1 nM or 3 nM PMP (Seahorse Biosciences), for either BMDMs or hepatocytes. State III respiration, defined as ADP-stimulated respiration in intact, unpoisoned mitochondria, in the presence of excess substrate was assessed[66].

Pyruvate elicited respiration was assessed in permeabilized cells offered 5 mM pyruvate, 2.5 mM malate (port A) and 0.2 μM rotenone (port B)[6,65]. The concentration of the aforementioned substrates are meant as final concentrations in wells after drug/substrate injections.

For Electron Flow Assay experiments, to allow examination of sequential electron flow through different complexes of the electron transport chain, isolated liver mitochondria were diluted in cold 1X MAS + substrate (10 mM pyruvate, 2 mM malate, and 4uM FCCP), to the concentration required for plating 3 μg mitochondria/well. 25 μL of mitochondrial suspension was aliquoted to each well of Seahorse XF96 Cell Culture Microplate and spun at 2000 × g for 20 min at 4 °C. After centrifugation, 155 μL of prewarmed (37 °C) 1x MAS + substrate was added to each well, the plate was then transferred to the Agilent Seahorse XFe/XF96 Analyzer and the run initiated. 2 μM rotenone was injected in port A, 10 mM succinate in port B, 4 μM antimycin A in port C and 10 mM ascorbate/100 μM TMPD in port D[86].

CI, CII activities were calculated subtracting respectively to basal OCR rotenone-elicited values, to succinate OCR antimycin A-elicited values; for CIV activity maximal values after ascorbate/TMPD were considered.

Modification of this protocol were made when other CI sources were tested; in this case isolated mitochondria were diluted in cold 1X MAS + 10 mM glutamate (in the presence or absence of 2 mM malate), and 4 μM FCCP[87].

## Metabolomics

WT and *Nos2*[−/−] BMDM (-10[7] cells per sample) were treated with vehicle (PBS) or LPS + IFNγ for 16 h. Cells were then washed, gently scraped, pelleted, snap frozen in liquid nitrogen and sent to Metabolon, Inc for metabolomic analyses.

Quantification of metabolites from lipoate treated cells was performed in house with high resolution qTOF ESI-LC/MS as previously described[88].

## 13C tracing studies

Leucine tracing buffer was made by adding dialyzed 10% FBS (Gibco), L-glutamine (1 mM) and L-Leucine U13 C6 (1 mM) (Cambridge Isotope Laboratories) to RPMI-1640.

Cells were activated for 16 h with LPS + IFNγ in RPMI media and then washed with PBS and incubated in tracing buffer at 37 °C for 4 h (confirmation of steady-state). After incubation cells were washed, extracted and analyzed in house as described above.

Isotopologue analysis of acquired negative and positive MS data was performed with MassHunter Profinder 8.0, which performed simultaneous data processing and quantitation of $^{13}C$ incorporation in a list of target metabolites derived from an in-house AMRT metabolite standard library. Extracted isotopologue abundances were corrected for their natural isotope abundance and corrected abundance percentages were calculated[6,88,89]. $M + 0$ to $M + n$ indicate the different mass isotopologues for a given metabolite with n carbons, where mass increases due to $^{13}C$-labeling.

## Reactions of NO reduction

To analyze the products of NO reduction in the presence of GSH[90], 0.25 or 0.5 mM DEA/NO were incubated with 0.25 or 0.4 mM of either reduced (Sigma, # T8260) or oxidized (Sigma, # T1395) lipoate for 1 h at 37 °C with different concentrations of GSH (50 or 100 μM). Reactions were conducted in 100 μL total volumes in 96 well plates. Vehicle controls were used in DEA/NO alone with GSH reactions as baseline. As positive control of HNO-derived conjugation with GSH, AS was used at

0.1, 0.5 or 1 mM. After the indicated reaction time reaction mixtures were directly analyzed by ESI-LC/MS-MS. To evaluate any contribution of DLD in NO reduction 7 μg of rhDLD was added to reaction mixtures.

## Direct HNO detection

BMDMs were treated with 12.5 μM HNO probe[56] for 30 min, then washed, and either left untreated or stimulated with LPS + IFNγ for 4 h. Additional conditions were represented by treatments with AS or DEA/NO (400 μM) for 30 min incubations. Cells were lysed in RIPA buffer and HNO-induced fluorescence (excitation 465 nm/emission 520 nm) was measured. DMSO vehicle controls were used as fluorescence background.

## HNO-derived metabolite quantification

Targeted measurements on crude reaction mixtures were performed through electrospray ionization mass spectrometry (ESI-LC−MS/MS) analysis in multiple reaction monitoring mode with an Agilent 6410B Triple Quadrupole mass spectrometer interfaced with a 1200 Series HPLC quaternary pump (Agilent) available in house at NCI-Frederick. Chromatographic resolution was obtained in reverse phase on a SB C18 column (1.8 μm; Agilent) with a flow rate set at 0.4 mL/min. The MRM transitions in the positive ion mode were $m/z$ 308.09 > 179 for GSH, $m/z$ 613 > 355 for GSSG, $m/z$ 337 > 307/231.9 for GSNO, $m/z$ 339.2 > 322 for GSONH2 and $m/z$ 340.2 > 322 for GSO2H[90]. Reaction mixtures containing rhDLD were mixed with 100% methanol, precipitated, dried using a vacuum concentrator and resuspended in milli-Q for analysis.

For detection of HNO metabolites in primary hepatocytes, cell pellets ($0.5 × 10^6$ cells) were washed and resuspended in 80% methanol. Phase separation was achieved by centrifugation at 4 °C and the methanol-water phase containing polar metabolites was separated and dried using a vacuum concentrator. The dried metabolite samples were stored at −80 °C and resuspended in milli-Q water the day of analysis.

## Immunoaffinity purification

DLAT and DLD (respectively PDH-E2 and E3) immunoaffinity purifications (IPs) from BMDMs for proteomic studies were performed using protein G Agarose resin (Pierce, #20398) coupled with 2μg of Anti-Pyruvate Dehydrogenase E2 antibody (Abcam, #ab172617) or Anti-Lipoamide Dehydrogenase antibody (Abcam, #ab133551). $7 × 10^7$ cells were resuspended in 450 μl Lysis Buffer (50 mM Tris pH 7.2, 300 mM NaCl, 0.5% NP-40 with protease and phosphatase inhibitors). Lysed cells were vortexed three times for 20 s each during incubation for 10 min at 4 °C. Insoluble material (pellet) was removed by centrifugation at 20,000 × g for 15 min at 4 °C. The supernatant was collected and DLAT or DLD were immunoisolated by incubation with pre-associated resin and antibodies for 2 h at 4 °C mixed by rotation in a final volume of 1 mL. The resin containing protein complexes was then washed three times with RIPA buffer and incubated with 2X SDS non-reducing Sample Buffer for 10 min at 70 °C.

## Isolation of mitochondria

For experiment in cultured cells, cellular fractionation into cytosol and intact mitochondria was done essentially as previously described[91,92]. Briefly, mitochondria from BMDM pellets (-10[7] cells) were isolated from the cytosolic fractions after cell permeabilization with a buffer containing 0.1% digitonin in 210 mM mannitol, 20 mM sucrose and 4 mM HEPES. The pellets after centrifugation at 700 × g for 5 min contained mitochondria.

Further differential centrifugation at 100,000 × g of these mitochondrial-enriched lysates was used to purify the mitochondrial matrix fraction. Swelling of the mitochondrial outer membrane was performed by hypotonic shock, followed by sonication (four repeats of 5 s- bursts at high settings with 30 s pauses), supplementation with

150 mM NaCl and 5% glycerol and centrifugation at $20,000 \times g$ 30 min. The supernatant was saved as soluble mitochondrial matrix fraction.

For isolation of mitochondria from whole liver, homogenization was performed as previously described[91,92] with glass–Teflon potter for 4–5 strokes. Total protein (mg/mL) was determined using the Bradford Assay reagent (Thermo). Mitochondria were resuspended in MAS IX buffer and incubated 1 h at 25 °C with 500 µM Angeli's salt or DEA/NO in gentle agitation. Then mitochondria were pelleted by centrifugation at $7000 \times g$, 10 min, 4 °C and used for subsequent assays.

### Native PAGE (BN-PAGE) and native immunoblots
The NativePAGE Novex Bis-Tris gel system (Thermo Fisher Scientific) was used for the analysis of native proteins as performed as in ref. [92]; 30 µg of mitochondrial matrix lysates were loaded/well; the electrophoresis was performed at 150 V for 1 h and 250 V for 1 h on ice. Where indicated, lysates were incubated additionally with DTT (10–100 mM, Sigma) for 1 h at 37 °C before electrophoresis. For the native IB, PVDF was used as the blotting membrane. The transfer was performed at 25 V for 1 h at 4 °C. After transfer, the membrane was washed with 8% acetic acid for 20 min to fix the proteins, and then rinsed with water before air-drying. The dried membrane was washed 5–6 times with methanol (to remove residual Coomassie Blue G-250), rinsed with water and then blocked for 1 h at room temperature in 5% milk, before incubating with the desired antibodies diluted in 2.5% milk O.N. at 4 °C. In order to avoid strip and reprobing of the same membrane, which might allow detection of signals from the previous IBs, samples were loaded and run in replicates on adjacent wells of the same gel, and probed independently with different antibodies.

### DLD in gel activity assay (diaphorase)
30 µg of mitochondrial matrix lysates, 0.25 µg of rhDLD or 30 µg of whole cell HEK293T lysates were run on Blue Native PAGE at 150 V for 1 h, followed by 1 h at 250 V on ice. Where indicated, lysates were incubated additionally with either DETA/NO (500 µM)/DTT/Lipoate (1 mM, Sigma) for 1 h at 37 °C before electrophoresis. The gel was soaked into 50 mM potassium phosphate buffer (pH 7.0) assay buffer, containing 0.2 mg/ml NBT and 0.1 mg/ml NADH for 30–45 min until DLD activity developed, according to a previously published protocol.

### DLD dehydrogenase activity
30 µg of mitochondria or cell derived mitochondria-enriched lysates, 0.1 µg of rhDLD (incubated with increasing concentration of AS or DEA/NO for 30 min at 37 °C) or 30 µg of whole cell HEK293T lysates were used to measure DLD activity by depletion of NADH. Where indicated, whole mitochondria or cell-derived mitochondria-enriched pellets were incubated additionally with either Cysteine/GSH/DTT (10 mM, Sigma)[52] in MAS buffer for 1 h at 37 °C before lysis. The reaction mixture contained 0.1 mM NAD, 0.2 mM NADH and 1 mM lipoic acid[93]. A baseline was determined through a sample containing the reaction mixture and enzyme diluting buffer. Spectrophotometric readings at 340 nm were measured every 20 s for 10 min. For kinetics studies non-linear regression of activity curves was calculated for Michealis–Menten model and linear regression of reciprocal axes to generate Lineweaver-Burk plot.

### OGDC activity
30 µg of cell derived mitochondria-enriched lysates were used to measure α-ketoglutarate dependent NADH production by monitoring NADH absorbance at 340 nm, according to a previously established method[94].

### Cytokine assay
The mouse cytokine array kit (R&D Systems, ARY006) was utilized for the parallel determination of the selected mouse cytokines levels produced by BMDMs. After treatments, cell culture supernatant was centrifuged to eliminate any suspended cells and 1 ml was used according to the manufacturer's protocol.

## Proteomics
**DDA analysis on immunoprecipitated DLAT.** Gel slice (cut at ~70 kDa) from SDS-PAGE of immunoprecipitated DLAT was dried with acetonitrile, washed, incubated with 20 mM iodoacetamide (IAM) for 1 h at 25 °C in the dark, treated for 1 h on ice with 10 ng/µL GluC[95] in 50 mM HEPES pH 8 and further incubated with 50 mM HEPES pH 8 at 37 °C overnight. Supernatant was removed and slice acidified to pH ~2 with TFA, then desalted on Oasis HLB µ-elution plate, eluted and dried in speedvac. Peptides were analyzed on an Easy-nLC 1200 system in front of a Orbitrap Fusion (Thermo) equipped with a Flex ion source. Solvent A consisted of 0.1%FA in water and Solvent B of 0.1%FA in 80%ACN. Samples were loaded onto the trap column (Acclaim™ PepMap™ 100 C18 HPLC Column, 3 µm, 75 µm I.D., 2 cm, Thermo PN 164535) and separated on the analytical column (Acclaim™ PepMap™ 100 C18 HPLC Column, 3 µm, 75 µm I.D., 25 cm, Thermo PN 164569). Samples were acquired using the TopSpeed method with a 3 s cycle time that consisted of the following: Spray voltage was 1700V and ion transfer temperature of 300 °C. MS1 scans were acquired in the Orbitrap with resolution of 120,000, $m/z$ range of 380–1550, automatic gain control (AGC) of $1e^6$ ions; MS2 scans were acquired in the Orbitrap using 30,000 resolution, AGC of $5e^4$. Data was searched in Proteome Discoverer using the most recent version of the Uniprot reviewed murine database, 2 max missed cleavages, peptide length of 6–60 amino acids, an MS1 mass tolerance of 10 ppm, and MS2 mass tolerance of 0.02 Da. Fixed modification of carbamidomethyl (+57.021 Da) on cysteine was included.

**Synthetic peptides with AS**. The synthetic peptide SGVENPGGYCL (1 µg) was treated with 20 mM AS (200 µM final) for 35 min at 37 °C. Samples were treated with 10% TFA, desalted and dried in speedvac. Dried peptide was resuspended in 0.5%FA and loaded for LC/MS analysis. All samples were analyzed in triplicate on an Easy-nLC 1200 system in front of a Q-Exactive (Thermo) equipped with a Flex ion source. Samples were acquired using the Top5 ions for MS2 method. Data processing was carried out in Skyline 21.1, using the sum of MS1 isotopic peak areas.

**Label-free quantification (LFQ) and tandem mass tag (TMT) analysis of rhDLD**. rhDLD (0.5 µg/condition) in 25 mM HEPES pH 7.3 was treated with 10 mM NaOH or either 200 µM AS, 200 µM DEA/NO, 200 µM $NaNO_2$ (all in 10 mM NaOH), with or without 200 µM GSH. All reactions were incubated at 37 °C for 35 min then HEPES pH 8, TCEP and IAM were added and incubated at 25 °C in the dark for 30 min. Then samples were digested with trypsin/LysC (#Thermo A40009) and each sample underwent TMTpro labeling (#Thermo A52045), then quenching of excess TMTpro before samples were combined together. Samples were cleaned and dried in speedvac. For LFQ analysis, FA was used to acidify the samples, which were cleaned and dried in speedvac. All samples were analyzed on an Dionex U3000 RSLC in front of a Orbitrap Eclipse (Thermo) equipped with an EasySpray ion source with FAIMS™ interface. For TMTpro samples, the dried peptides were resuspended in 50 µL 0.1% FA and 15 µL was injected three times, each using a method with different FAIMS compensation voltage (CV) combinations. MS1 scans were acquired in the Orbitrap with resolution of 120,000, AGC of $4e^5$ ions, mass range of 350-1600 m/z; MS2 scans were acquired in the Orbitrap using TurboTMT method with resolution of 15,000, AGC of $1.25e^5$, intensity threshold of 2.5e4 and charges 2–6 for MS2 selection. The only difference in the methods was the CVs used (method 1 used −45, −60, −75; method 2 used −50, −65, −80, and method 3 used −55, −70, −85). For label free samples, the dried peptides were resuspended in 50 µL 0.1% FA and 10 µL was injected once using a method consisting four FAIMS CVs (−45, −55, −70, −85). MS1 scans

were acquired in the Orbitrap with resolution of 120,000, AGC of $4e^5$ ions; MS2 scans were acquired in the Ion Trap using rapid acquisition, AGC of $1e^5$, intensity threshold of 1e4 for MS2 selection. Data processing was carried out in the most recent version of Skyline, using the sum of MS1 isotopic peak areas. All data were searched in Proteome Discoverer 2.4 using the Sequest node. Both datasets were searched against the Uniprot Mouse database from Feb 2020 using a full tryptic digest, 2 max missed cleavages, peptide length of 6–40 (TMTpro) and 6-60 (LFQ) amino acids, an MS1 mass tolerance of 10 ppm, and MS2 mass tolerance of 0.02 Da. For both TMTpro and LFQ datasets the following variable modifications were included with the search: oxidation on methionine (+15.995 Da), modification of carbamidomethyl (+57.021 Da), trioxidation/sulfonic acid (+47.985 Da), sulfinic acid (+31.990), and Sulfinamide (+31.005) on cysteine. For TMTpro data variable modification of TMTpro (+304.207 Da) was included for lysine residues and peptide N-terminal amines. TMTpro reporter ions were quantified using the Reporter Ion Quantifier node and LFQ data was quantified using precursor abundance intensity. Both TMTpro and LFQ data were normalized on total peptide abundance.

**PRM analysis on DLD Cys484-peptide.** rhDLD in 25 mM HEPES pH 7.3 was treated with 10 mM NaOH, or 200/400/800 μM AS, or 200/400/800 μM DEA/NO. All reactions were incubated at 37 °C for 35 min then HEPES pH 8 with TCEP and IAM were added and incubated at 25 °C in the dark for 30 min followed by trypsin/LysC at 37 °C for 3.5 h. Samples were digested with trypsin and dried in speedvac. For analysis on mitochondria enriched lysates, proteins were precipitated, resuspended in EasyPep lysis buffer and treated with reducing and alkylating solutions and then heated at 95 °C for 10 min. Samples were digested with trypsin/LysC and dried in speedvac. The dried 50 μL aliquot was resuspended in water and peptide concentration was determined using the Pierce™ Quantitative Colorimetric Peptide Assay. Equal amounts of peptides were resuspended in the heavy isotope peptide mixture directly. All samples were analyzed in triplicate on an Easy-nLC 1200 system in front of a Q-Exactive HF (Thermo) equipped with a Flex ion source. Samples were acquired using targeted parallel reaction monitoring (PRM) for the both endogenous and heavy isotopes (on the C-terminal K or R) for the following DLD peptides: VVHVNGYGK (charge +2, $m/z = 486.767/490.774$, HCD = 27%), GRIPVNTR (charge +2, $m/z = 465.772/461.776$, HCD = 35%), EANLAASFGK (charge +2, $m/z = 504.262/508.269$, HCD = 27%), VC[+57]HAHPTLSEAFR (charge +3, $m/z = 508.917/512.293$, HCD = 27%), VC[+48]HAHPTLSEAFR (charge +3, $m/z = 505.905/509.241$, HCD = 27%), VC[+32]HAHPTLSEAFR (charge +3, $m/z = 500.574/5031.910$, HCD = 27%). Data processing was carried out in Skyline, using the sum of fragment ion areas for each peptide PRM. The ratio of Light to Heavy (L/H) for each peptide was calculated, then the ratio of ratios was calculated for modified peptides versus the VVHVNGYGK peptide to account for differences in total DLD amounts.

**TMT – global mitochondrial proteomic analysis.** Proteins from mitochondria-enriched lysates (see first step of mitochondria isolation from cultured cells above) were precipitated, resuspended in EasyPep lysis buffer and probe sonicated to aid in solubilization. Samples were treated with reducing and alkylating solution provided with EasyPep kit and digested with trypsinc/LysC. 25 μg for each sample was treated with 30 μL of 5 μg/μL TMTpro label and incubated at 25 °C for 1 h followed by quenching before samples were combined together. The combined TMTpro labeled peptides were desalted and dried in speedvac. Aliquots of TMTpro labeled peptides were analyzed an Dionex U3000 RSLC in front of a Orbitrap Eclipse (Thermo) equipped with an EasySpray ion source with FAIMS™ interface. All data were searched in Proteome Discoverer 2.4 using the Sequest node. Both datasets were searched against the Uniprot Mouse database from Feb 2020 using a full tryptic digest, two max missed cleavages, peptide

length of 6-144 amino acids, an MS1 mass tolerance of 10 ppm, MS2 mass tolerance of 0.02 Da, fixed TMTpro modification (304.207 Da) on lysine and N-terminal amines, variable oxidation on methionine (+15.995 Da), and the following variable modification on cysteine: carbamidomethyl (+57.021 Da), trioxidation/sulfonic acid (+47.985 Da), sulfinic acid (+31.990), Sulfinamide (+31.005), and TCEP modification (+266.208). TMTpro reporter ions were quantified using the Reporter Ion Quantifier node and normalized on total peptide amount. Sum of peptides/ protein and normalized Cys-modified peptide abundances were converted to a Log2 scale, and biological replicates were grouped by experimental condition. Peptide modification was normalized for protein abundance, providing the normalized modification level for each cysteine-containing peptide in the analyzed proteome (cutoff log2FC > 0.15 of stimulated vs untreated). Data were visualized using a Volcano plot, which shows the log10 fold change on the x-axis and the adjusted $p$ value on the y-axis. Intersection with proteins in DLD interactome was performed interrogating UniProt database at https://www.ebi.ac.uk/intact/search?query=id:P09622*#interactor. Over-representation analysis of significant changes were assessed using Metascape[96], GO[97–99] and Enrichr[100].

**TMT - free cysteine analysis.** Mitochondria enriched lysates were treated with 15.3 mM Iodoacetyl-PEG2-Biotin (#Thermo 21334) to label free cysteine residues and incubated for 18 h with shaking at 25 °C in the dark. Proteins were precipitated and resuspended in EasyPep lysis buffer (Thermo #A40006, A45734, A45733) and treated with reducing and alkylating solution. Then trypsin/Lys-C digestion was performed and each sample was incubated with TMTpro label at 25 °C for 1 h. Excess TMTpro was quenched before samples were combined together. The mixed TMTpro sample was desalted and dried in speedvac. Aliquots were combined and enriched using High-Capacity Streptavidin Agarose slurry (50% aqueous slurry, 200 μg of beads)(# Thermo 20537) following the manufacturer's protocol. The bound peptides were eluted with 50%ACN, 0.5%TFA and the eluted fractions were combined and dried. TMTpro labeled Global peptide and streptavidin enriched IAM-PEG2 peptides were analyzed on an Dionex U3000 RSLC in front of a Orbitrap Eclipse (Thermo) equipped with an EasySpray ion source with FAIMS™ interface. All data were searched in Proteome Discoverer 2.4 using the Sequest node. Both datasets were searched against the Uniprot Mouse database from Feb 2020 using a full tryptic digest, two max missed cleavages, peptide length of 6–60 amino acids, an MS1 mass tolerance of 10 ppm, MS2 mass tolerance of 0.02 Da, fixed TMTpro modification (+304.207 Da) on lysine and N-terminal amines, variable oxidation on methionine (+15.995 Da), and variable modification of carbamidomethyl (+57.021 Da) and IAM-PEG2-Biotin (+414.194 Da) on cysteine. TMTpro reporter ions were quantified using the Reporter Ion Quantifier node and normalized on total peptide amount.

**Molecular modeling investigations**
PyMOL was used to build structural models of the human dihydrolipoamide dehydrogenase (DLD) modified at Cys484 or at Cys477 by adding a sulfinamide functional group in place of the native thiol functional group, by using as a protein template the crystallized structure of the human DLD (also known as E3, PDB_ID: 3rnm). DLD_E3 reported in 3rnm.pdb was solved as a FAD containing DLD_E3 homodimer in complex with the subunit binding of human dihydrolipoamide transacylase (E2b)[101]. The sulfinamide functional groups was introduced at Cys484 or at Cys477 of both monomers of 3rnm.pdb. In order to investigate the role played by C484 in protein function, in silico site directed mutagenesis was performed for building C484A and C484W single mutants by using the obtained 3D models and the mutagenesis tool implemented in PyMOL.

For docking NAD+ cofactor in DLD-NAD+ binding region within 3rnm.pdb entry, the crystallized structure of another human DLD_E3

crystallized in complex with FAD and NAD$^+$ (1zmd.pdb)[59] was superposed on 3rnm.pdb. Thus, a NAD+ object was duplicated from 1zmd.pdb superimposed to 3rnm.pdb and saved within the aligned DLD-NAD$^+$ binding region of the 3rnm.pdb entry, by using the super command from PyMOL. The "super" command allows aligning the selected proteins under investigation for performing a comparative structural analysis, due to its ability in providing a sequence-independent structure-based pairwise alignment, more efficient than the sequence based pairwise alignment obtained by using the "align" command[74,102].

All the generated 3D all-atom models were optimized by using the FOLDX repair tool[103] and then energetically minimized using the Yasara Minimization server[104] or the Rosetta "relax" application within the Rosetta "scoring and prep" tools[105,106], independently. Relaxation as the energy minimization process used to optimize protein structures involves iteratively adjusting the conformation of the atoms in the protein to find a low-energy state. This step is crucial for refining and improving protein structures obtained from experimental data or generated through modeling approaches. It helps to relieve steric clashes, correct small-scale structural errors, and optimize side-chain conformations, ultimately resulting in energetically favorable and more accurate protein models. The obtained final models were examined in VMD, PyMOL, and SPDBV by visual inspection searching for putative unsolved clashes[74,75,102,107].

The FoldX AnalyseComplex assay was performed to determine the interaction energy at the interface between the two DLD-E3 monomers or at the interface between the two DLD-E3 monomers and the E2b subunit, both in the wild-type crystallized 3rnm.pdb structure and in the Cys$^{484}$-sulfinamide modified protein, or in the Cys$^{477}$-sulfinamide modified protein for comparative purposes. The interaction energy at the protein-protein interface of the investigated mutants C484A and C484W was also calculated by using the same FoldX AnalyseComplex assay.

The way the FoldX AnalyseComplex operates is by unfolding the selected targets and determining the stability of the remaining molecules and then subtracting the sum of the individual energies from global energy. More negative energies indicate a better binding. Positive energies indicate no binding[103,108]. The interaction energies calculated for the crystallized structure at the monomer-monomer interface within the DLD_E3 dimer or at the dimer-monomer interface within the DLD_E3-E2b trimer were used as a reference value. "InterfaceAnalyzer" application from Rosetta "scoring and prep" tools was used was used for evaluating interactions at the monomer-monomer interface within the DLD_E3 dimer or at the dimer-monomer interface within the DLD_E3-E2b trimer both in the DLD_E3-E2b wild type and in the DLD_E3-E2b_C484 sulfinamide modified protein complex[106,109]. Furthermore, the "residue_energy_breakdown"[109,110] application was also used to examine the contribution of each protein residue to the score for the lowest energy backbone relaxed structures in the investigated protein complexes (WT.vs.C484sulfinamide modified protein; WT.vs.C477sulfinamide modified protein).

### Transfection of HEK293T cells
Empty vector (pCMV6-Entry) and WT human DLD Tagged ORF Clone were purchased from Origene (#PS100001 and #RC200639), DLD C484A and C484W mutant Tagged ORF Clones were custom synthesized through GenScript.

HEK293T cells ($2 \times 10^5$) were seeded in 10 cm$^2$ tissue culture dishes. The next day, cells were transfected using Lipofectamine 3000 reagent (Invitrogen) according to the manufacturer's instructions. Briefly, 25 μg plasmid DNA was mixed with 600 μl serum free medium + 17 μl P3000 reagent. The solution containing DNA was gently mixed with 600 μl serum free medium containing 25 μl Lipofectamine and incubated at room temperature for 20 min. The entire DNA:Lipofectamine complexes were added dropwise onto the cells and mixed by gentle swirling. After 24 h, the transfected cells were used for downstream applications. Whole cell lysates for native gels were generated by adding a buffer containing 50 mM Bis-Tris, 50 mM NaCl, 10% w/v Glycerol, 0.001% Ponceau S, 1% lauryl maltoside, pH 7.2 and protease inhibitors to intact cells followed by incubation of 10 min in ice and centrifugation at 20,000 × g at 4 °C for 20 min.

### Statistical analysis
Data are presented as means ± standard error of mean and are representative of at least two independent experiments. Statistical analysis was performed using GraphPad Prism 9. One-way or two-way ANOVAs followed post tests and linear regression were used when multiple groups were analyzed as are indicated in figure legends. Student's $t$ test was used for statistical analysis when two groups were analyzed. All data are assumed normally distributed and were log transformed where appropriate before analysis. Statistical significance was set at *$p < 0.05$. Exact p values for significant comparisons are reported in the figures. P values > 0.05 are reported as "ns" unless otherwise specified.

### Reporting summary
Further information on research design is available in the Nature Portfolio Reporting Summary linked to this article.

## Data availability
All data supporting the findings of this study are available with the article, and can also be obtained from the corresponding author. Publicly available crystallized structure of the human DLD (PDB_ID: 3rnm) and hDLD crystallized in complex with FAD and NAD$^+$ (PDB ID: 1zmd) were used in this study. The proteomics data have been deposited in MASSive under accession code MSV000092336. Source data are provided with this paper.

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

## Acknowledgements

This research was supported, in part, by the intramural Research Program of the NIH, National Cancer Institute USA. Structural analyses were supported by Italian Association for Mitochondrial Research resources made available by ReCaS, a project funded by the MIUR (Italian Ministry for Education, University and Research) in the "PON Ricerca e Competitività 2007–2013-Azione I-Interventi di rafforzamento strutturale" PONa3_00052, Avviso 254/Ric, University of Bari ("Fondi Ateneo ex-60%"2016"; "ProgettoCompetitivo 2018" and "FFABR 2017–2018") (to C.L.P. and V.T.). The content of this publication does not reflect views or policies of the Department of Health and Human Services, nor does mention of trade names, commercial products, or organization simply endorsement by the U.S. Government. Illustrations in Figs. 1d, 2a,j, 3e, 4a, 7f, 8, S1a, S2a,h, S3a, S5d,e, S6a,e were created with BioRender.com.

## Author contributions

E.M.P. conceived and performed experiments, and wrote the manuscript. R.H., N.M., and J.W. performed experiments. C.M verified chemical methods and conditions. D.W.M helped design studies, wrote the manuscript and secured funding. C.L.P. and V.T. conducted structural analysis. R.H. acquired and analyzed proteomics data. D.A.W, T.A.R, T.A., and K.M.M. provided reagents, expertise, and feedback.

## Funding

## Competing interests

The authors declare no competing interests.
