## [Peer Review File · Nature Communications]

REVIEWER COMMENTS

Reviewer #1 (expert in macrophage immunometabolism):

Palmieri et al. show that LPS/IFN γ -induced downregulation of mitochondrial respiration and PDH activity can be (partly) assigned to NO production and related post-translational modifications of PDH enzymes and metabolic rewiring. With further molecular analysis of proteins and their modifications, they find modified cysteine residues that affect DLD activity. The manuscript tells a comprehensive and complete molecular story and adds to the macrophage-metabolism field by providing the importance of other post-translational modifications for macrophage metabolic rewiring beyond acetylation and phosphorylation. However, the manuscript and its reasoning are sometimes a bit hard to follow or under-discussed and lack some functional application in macrophages, as discussed in the points discussed below.

1. First, because of the many different enzymes, metabolites and cysteines described and explored, the manuscript would benefit from a graphical abstract showing the (possible) molecular mechanisms that are studied.
2. In the first figure, NO is appointed to be causative for LPS/IFN γ -mediated reduced PDH activity via the downregulation of lipoate levels. However, figure 1a shows an even further reduced PDH activity in LPS/IFN γ macrophages than in macrophages treated with DETA/NO, indicating that there might be (an) additional parallel mechanism that has not been discussed by the authors.
3. In the discussion, the authors speculate on HNO being a possible therapeutic target for intervention. However, the manuscript lacks any functional description of the mechanisms in the macrophage and the reliance of their inflammatory phenotypes on the described pathways. It would be valuable to include some functional assays that measure the inflammatory capacity of the cells for example in response to AS and give a direction for possible future therapeutic intervention.
4. Are GS(O)NH₂, GSN(O)H₂, and GSH-sulfonamide the same molecule? If yes, please be consistent in their naming throughout the text and the figures. If no, please specify the difference a bit clearer to make the figure and the text more comprehensible in the paragraph on 'HNO is generated by NO in the presence of reduced lipoate and is detectable in BMDMs'.
5. Similarly, the mixed use of PDH-E3 and DLD, M1 and 'stimulated macrophage' throughout the manuscript is confusing and would benefit from sticking to one name.
6. The authors performed extensive proteomic analysis and revealed PTMs on proteins belonging to certain pathways. However, they barely speculate on the direction of regulation (Positive or negative) on the regulation of these pathways.
7. The authors have used overnight (O.N.) stimulation throughout the manuscript. The term overnight stimulation is a bit vague and not specified in the manuscript. As it can range from roughly 12-20 hours, and it is compared in some figures to 8-hour stimulation, a specific time would be required for better interpretation of the data.

Minor comments

8. There is no consistency in the abbreviations used in axis labels (e.g. LPS+IFN γ is sometimes abbreviated to Ly and not clearly explained in the legend)
9. Statistics are missing in figures 1A and 2D although differences are stated in the text.
10. Axis of Fig S5A is not readable in the current format.
11. Fig S5b is not referred to in the text
12. Fig1F-H: it is unclear what the number of tested samples is.

Reviewer #2 (expert in metabolomics):

HNO plays significant roles in many biological processes, such as vascular relaxation, enzyme activity regulation, neurological function regulation, and anti-inflammatory effects. In this study, the authors found that lipoate can facilitate NO-mediated HNO production for modifications of PDH complex. This is an interesting and novel finding that lipoic acid and HNO chemistry serve as the factors for modifications of PDH complex. However, I suggest that the authors should design experiments for better understanding how lipoic acid facilitates the HNO binding. Moreover, next questions should be addressed before the manuscript might be accepted for publication in Nature

Communications.

(1) It should be noted that the authors demonstrated that NO targets aconitase 2 (ACO2) and pyruvate dehydrogenase in macrophages in previous report (please refer to PMID: 32019928). Therefore, I am interested whether HNO-binding ACO2 for regulating macrophage polarization.

(2) The details of the endogenous generation of HNO are still unclear. Whether and how lipoate promote HNO generation in the cells?

(3) Since the addition of lipoyl modifications to substrates has been best characterized via two independent pathways (please refer to PMID: 29169048), either environmentally-obtained lipoic acid or de novo lipoic acid synthesis in mammals can support protein lipoylation. Therefore, whether the authors identified lipoylation of mitochondrial proteins including PDH complex by adding lipoic acid (200 μ M) as well as modulating the de novo pathway in cell culture? (please refer to PMID: 33174356, PMID: 34581824)

(4) The abundant heme iron centers in mitochondria are available for the formation of HNO via the reaction between NO and H₂S. Regarding the functions of modifications of PDH complex, whether and how PDH regulates iron oxidation state after E2/E3 modified by HNO (please refer to PMID: 27797441).

(5) The authors stated that depletion of cellular GSH with BSO, pushing the equilibrium of HNO chemistry towards sulfinamide (Fig. 2b). They should further use NAC (N-acetylcysteine) for GSH generation to reduce sulfinamide in the validation tests.

Minor comments:

(1) The reference format should be corrected by using abbreviation of the journal names.

(2) Metabolomics also support the idea that BCKDH, OGDH and GLDC are targeted in a NO-dependent manner (Fig. S9d). I suggest that they further perform stable isotope tracing analysis for validation of BCKDH, OGDH and GLDC activities.

Reviewer #3 (expert in nitric oxide signaling and gas sensing):

This paper is not an easy read, both from the perspective of how it is written and the scope of the experiments and the relation of those experiments to the premise put forward and the conclusions reached. The recommendation is to reject the paper as not being suitable for Nature Communications. With revision, it would certainly be appropriate for a more specialized biochemical journal but it lacks the breadth of discovery most appropriate for this journal.

General comments:

In the abstract and in various places throughout the paper the authors refer to HNO directly targeting the PDH complex. This is misleading. More appropriate it to describe this as a facile chemical reaction, likely due to the proximity of generation.

The following is one example of an unclear sentence taken from the abstract (underlining mine). A thorough grammatical review by the authors is recommended. "Mechanistically, we show that lipoate facilitates NO-mediated HNO production that interacts with thiols forming modifications resistant to reduction".

The authors state (line 43) this HNO formation is "peculiar to the lipoate rich PDH complex". I agree but this very description limits the breadth of the findings. It is a very good piece of mechanistic work on this specific reaction.

The "rewiring" concept is an overreach (line 62 and throughout). For example, how is this different from when a cell utilizes fatty acids under limiting glucose? It is an alternate pathway. Rewiring would involve more than just the use of an alternate pathway to generate energy.

Line 63 mentions "suppression" of the PDH complex. This is wrong word to describe what is happening and I think leads the reader astray. The PDH complex is inhibited and thus pyruvate concentrations increase with the attendant regulation by pyruvate taking over.

In lines 79-91, the authors summarize their work and others concerning how HNO might form and

aspects of reaction rates with other molecules but they conclude with speculation on HNO as a signaling agent. Nothing in the literature as yet outlines the key aspects of signaling for HNO (e.g. spatial and temporal control) but more importantly, this paper is not about signaling and does not add to arguments in favor of HNO signaling but instead is focused on the formation of HNO from lipoate in the E2 subunit and subsequent reactions. Again, referring to the reaction as "targeting" (line 89) is misleading. Same on line 96. Better wording would be 'reacts with'. It is more indicative of what is taking place.

Specific comments on results:

Beginning with line 107 the authors describe lipoate supplementation experiments. This is a complicated experiment. Scavenging of lipoate is pretty well worked out in bacteria so if we assume the same machinery exists in eukaryotic cells then, at a minimum, a lipoyltransferase would be required after activation of lipoate. Assuming these steps take place, how does PDH activity get revived? Is the complex disassembled and new E2 put in? Is the entire complex tagged for degradation and new PDH made? If the latter, why do you need exogenous lipoate?

The proteomic analysis (beginning on line 182) indicates rather broad reactivity of HNO. There is some focus on mitochondrial proteins but others not. The arguments about localization and reactivity seem weaker in light of these results.

The in vivo approximation experiments to find DLD Cys modifications (line 248 and following) are a concern. The sensitivity seems high enough for detection.

The in silico results reported on line 283 and following beg for experimental verification and are not compelling without such experiments.

RESPONSE TO REVIEWERS' COMMENTS

Reviewer 1.

Comment:

"The manuscript tells a comprehensive and complete molecular story and adds to the macrophage-metabolism field by providing the importance of other post-translational modifications for macrophage metabolic rewiring beyond acetylation and phosphorylation. However, the manuscript and its reasoning are sometimes a bit hard to follow or under-discussed and lack some functional application in macrophages, as discussed in the points discussed below.

1. First, because of the many different enzymes, metabolites and cysteines described and explored, the manuscript would benefit from a graphical abstract showing the (possible) molecular mechanisms that are studied. ."

Response:

We thank the reviewer for this suggestion, and we have now provided a more comprehensive graphical abstract and illustrations throughout the manuscript that depict the intramolecular generation of HNO and its role in suppressing activity of PDH, OGDH, and BCKDH.

Comment:

"2. In the first figure, NO is appointed to be causative for LPS/IFN γ -mediated reduced PDH activity via the downregulation of lipoate levels. However, figure 1a shows an even further reduced PDH activity in LPS/IFN γ macrophages than in macrophages treated with DETA/NO, indicating that there might be (an) additional parallel mechanism that has not been discussed by the authors. "

Response:

In this figure we assess total PDH activity to show that NO is sufficient to elicit decrease in total activity. In this experiment we have used a 16h treatment of 100 μ M DETA/NO; although it may appear based on this one concentration that DETA/NO does not lower PDH activity to the level of LPS+IFN γ macrophages, we know from our previous work (PMID: 32019928), and from this work (Fig. 7p) that in macrophages and hepatocytes, 500 μ M DETA/NO is the concentration that best recapitulates physiological conditions of the high intracellular NO associated with LPS+IFN γ . Based on this comment, throughout the manuscript we have now clarified in the figure legends the concentrations of DETA/NO used in the different experiments.

Comment:

"3. In the discussion, the authors speculate on HNO being a possible therapeutic target for intervention. However, the manuscript lacks any functional description of the mechanisms in the macrophage and the reliance of their inflammatory phenotypes on the described pathways. It would be valuable to include some functional assays that measure the inflammatory capacity of the cells for example in response to AS and give a direction for possible future therapeutic intervention."

Response:

We thank the reviewer for raising the important issue of how macrophage function might be affected specifically by HNO. We have now addressed this comment in three ways. First, to directly assess the impact of HNO on function, we tested the cell culture media from macrophages stimulated with LPS+IFN γ or those treated with the HNO donor. After assessing >40 soluble mediators, we found that cytokines classically associated with macrophage pro-inflammatory activation such as IL-6 and TNF α , were not affected by the presence of the HNO

donor. In contrast, we did detect consistent dysregulation in the production of CCL1, IL-1 α , C5a, GCSF, CXCL10, and KC (now Fig. S4I). Changes in the levels of these cytokines and chemokines can have significant effects on macrophage function, including their ability to migrate to sites of infection or inflammation, their phagocytic activity, their survival and proliferation and their ability to regulate the recruitment and activation of other immune cells (PMID: 29247991, PMID: 15771587, PMID: 30465827).

Second, we find that our proteomics analysis revealed a signature of irreversibly modified proteins associated with cell shape and cytoskeletal activities (Fig. 4d-e), which is consistent with our previous reports of NO-dependent morphological changes during M1 polarization (PMID: 31683257). This would suggest that since cellular morphology can be used as an index of inflammatory status, endogenous HNO could have a role in this phenotype.

Third, in our revised manuscript, we have extensively mined our proteomics datasets of irreversible modifications from stimulated macrophages and macrophages treated with the HNO-donor focusing on the relative total abundance of peptides across samples. This generates new proteomics data on overall total protein level changes in HNO donor treated cells vs control. We found 64 proteins to be significantly upregulated and 47 proteins downregulated in the presence of the HNO donor (using a cutoff of FC=1.25) (Fig. S4i). We were not surprised that this number is quite small considering the short time of treatment (4h). These data suggest that the addition of HNO alters the levels of proteins involved primarily in phagocytosis, intracellular transport and autophagy (now Fig. S4j-k). These are cellular pathways that play important roles in maintaining cellular homeostasis. In macrophages, these pathways are particularly important for regulating immune responses and ensuring efficient removal of cellular waste and pathogens. Retrograde transport is a pathway that allows for the transport of molecules and cargo from the Golgi apparatus and endosomes back to the endoplasmic reticulum (ER). Endosomal and retrograde transport play important role in maintaining the composition and function of the ER and are also involved in the regulation of immune responses (PMID: 24591520, PMID: 33397928), therefore it will be interesting to further dissect how HNO may regulate these phenomena.

Taken together, our cytokine/chemokine assessment and this new global proteomic information suggest that unlike NO, HNO may not be strictly involved in the classical inflammatory capacity of macrophages, but this specie may have distinct roles in regulating intracellular transport, ER homeostasis and cell migration which are important in tuning inflammatory status and perhaps directing response towards resolution. These results are now part of supplemental Figure 4 and explained at the highlighted lines 239-249 and 481-489 in the new version of the manuscript, and based on these results we have modified our speculation on possible therapeutic interventions.

Comment:

“4. Are GS(O)NH₂, GSN(O)H₂, and GSH-sulfinamide the same molecule? If yes, please be consistent in their naming throughout the text and the figures. If no, please specify the difference a bit clearer to make the figure and the text more comprehensible in the paragraph on ‘HNO is generated by NO in the presence of reduced lipate and is detectable in BMDMs’.

We apologize for our inconsistent nomenclature. We have now uniformly named GSH-sulfinamide throughout the manuscript after clarifying the formula when first mentioned.

Comment:

"5. Similarly, the mixed use of PDH-E3 and DLD, M1 and 'stimulated macrophage' throughout the manuscript is confusing and would benefit from sticking to one name"

Response:

We thank the reviewer for this suggestion. Considering that we now have added data on BCKDH in the main figures, we homogeneously refer to DLD throughout the manuscript, after introducing it as the E3 subunit of referred dehydrogenases. We have also specified in the methods that "activation", "stimulation" and "(M1 polarization" in the manuscript are referred to LPS + IFN γ treatment, unless otherwise specified.

Comment:

6. The authors performed extensive proteomic analysis and revealed PTMs on proteins belonging to certain pathways. However, they barely speculate on the direction of regulation (Positive or negative) on the regulation of these pathways.

Response:

Irreversible modifications of cysteine residues have been shown to regulate the activity, localization, and stability of proteins. These modifications provide a mechanism for fine-tuning cellular signaling pathways and can have important implications for various physiological and pathological processes. While the reversible cysteine post-translational modifications (PTMs) modulate protein function and act as redox switches in cellular signaling, modifications considered to be irreversible (e.g. to sulfinic or sulfonic acids) most often lead to protein denaturation and severe protein damage (PMID: 23514336). In other circumstances proteins are not degraded, but they are simply inactivated. For example, sulfinic acid modification (-SO₂H) has been shown to regulate the activity of many enzymes, including phosphatases and kinases. For example, sulfinic acid modification of the cysteine residue in the active site of protein-tyrosine phosphatases (PTPs) results in inhibition of PTP activity (PMID: 12802338). This modification is important for regulating cellular signaling pathways and has been implicated in various diseases, including cancer and diabetes (PMID: 15186772). More stable than sulfinic acid, sulfonic acid modification (-SO₃H) has been found to regulate the activity of several redox-sensitive proteins. One example is peroxiredoxin, which is a critical antioxidant enzyme that protects cells from oxidative stress. The sulfonic acid modification of a cysteine residue in peroxiredoxin is believed to serve as a mechanism for inactivating the enzyme when excessive oxidative stress occurs (PMID: 18725414).

Our data show that voltage-dependent anion selective channels (VDAC), the most abundant family of proteins in the mitochondrial outer membrane, are targeted by HNO-mediated irreversible modification in our cells (Fig. 4j-k). These proteins have been suggested to act as a sensor for mitochondrial damage. Over-oxidation of these proteins induces removal of damaged mitochondria. As such, modifications of VDACS and other transporters during HNO donor treatment may indicate increased turnover of damaged mitochondria in response to HNO to maintain a functional mitochondrial network (PMID: 28066720).

Nonetheless, we feel it would be overreaching to speculate whether the enriched pathways found in our system as irreversibly modified targets are being positively or negatively regulated by RNS and specifically by HNO (directly or indirectly). The most likely outcome is inhibition, but such a conclusion requires a thorough dissection of these pathways.

Comment:

7. The authors have used overnight (O.N.) stimulation throughout the manuscript. The term overnight stimulation is a bit vague and not specified in the manuscript. As it can range from roughly 12-20 hours, and it is

compared in some figures to 8-hour stimulation, a specific time would be required for better interpretation of the data.

Response:

Apologies for the confusion this caused. We have now revised our manuscript specifying the time of stimulation more precisely. Our overnight assays are for 16 hrs.

Minor Concerns

Comment:

"8. There is no consistence in the abbreviations used in axis labels (e.g. LPS+IFN γ is sometimes abbreviated to Ly and not clearly explained in the legend) "

Response:

We apologize for the lack of consistency in abbreviations. We have now revised figures and text using the consensus nomenclature of "LPS+IFN γ " for the combination of treatments.

Comment:

"9. Statistics are missing in figures 1A and 2D although differences are stated in the text"

Response:

We now provide statistical analysis of figure 1a and 2d.

Comment:

10. Axis of Fig S5A is not readable in the current format.

Response:

We thank the reviewer for pointing out this inaccuracy. We have now modified panel S5a with clearer and bigger x-axis.

Comment:

11. Fig S5b is not referred to in the text

Response:

We apologize for this oversight. We have now modified the text referring to Fig S5b.

Comment:

12. Fig1F-H: it is unclear what the number of tested samples is.

Response:

We thank the reviewer for raising this point. We have now included sample numbers in figure legends. Considering that the mass spectral data report area values and the magnitudes of these depend on daily instrument performance and changes in sample matrix effects between different experiments, we felt it wasn't proper to merge these data and report error bars in Fig. 1f-h. Since the trends observed were consistent across repeated experiments, we instead presented n=1 representative experiment. We have now noted this in the legend by declaring that presented data are representative of 3 experiments with similar results (n=3).

Reviewer 2.

Comment 1:

“This is an interesting and novel finding that lipoic acid and HNO chemistry serve as the factors for modifications of PDH complex. However, I suggest that the authors should design experiments for better understanding how lipoic acid facilitates the HNO binding.”

Response:

We regret that we were not more clear in the details regarding our model. HNO is highly reactive, binding to metals but also reacting with thiols at a high rate. Here, we are proposing that reaction of NO with the thiols on lipoate facilitates the in situ generation of HNO in close proximity to the DLD cysteines, resulting in rapid reaction of HNO with the DLD thiols. This results in irreversible modification of DLD, reducing its activity, likely via disruption of the dimer formation. The data demonstrate the ability of lipoate to facilitate HNO generation in vitro in the presence of NO. In fact, an HNO detector showed increased HNO in cells treated with LPS+IFN γ , in a manner that is sensitive to inhibition of iNOS. In addition, we find that LPS+IFN γ -induced HNO production in cells is substantially reduced by interfering with cellular lipoylation with a specific inhibitor. Lastly, the peptide modifications we detect are consistent with reaction of thiols with HNO. We have now clarified the text regarding this model, and we provide additional information below in response to comment 3.

Comment 2:

“(1) It should be noted that the authors demonstrated that NO targets aconitase 2 (ACO2) and pyruvate dehydrogenase in macrophages in previous report (please refer to PMID: 32019928). Therefore, I am interested whether HNO-binding ACO2 for regulating macrophage polarization. ”

Response:

While both HNO and NO bind to iron, in keeping with their redox relationship, HNO binds facily with ferric (Fe(III)) centers and reversibly with ferrous (Fe(II)) centers, while the reverse occurs for NO. Based on these chemical characteristics, the iron-sulfur cluster of ACO2 is expected to be more susceptible to NO-induced inhibition as opposed to inhibition by HNO. To address this reviewer’s comment, we have now measured ACO2 activity in BMDMs treated with different concentrations of DEA/NO (NO donor) and our HNO donor, and found that, contrary to PDH, ACO2 is promptly and strongly inhibited by increasing concentrations of NO (as we have previously shown), and less so by HNO (figure for reviewer 2 below). These findings support that the ACO2 iron sulfur cluster is a preferential target for NO rather than HNO. The minor inhibition of ACO2 induced by HNO, compared to strong inhibition by NO, may be attributed to HNO reactivity with small concentrations of an oxidized site. We thank the reviewer for this suggestion as the result has strengthened our model, suggesting some specificity of the intramolecular generation of HNO in PDH. (Please see Figure below for Reviewer 2)

Legend: Aconitase activity in mitochondrial lysates from WT BMDMs after treatment with either AS, DEA/NO or nitrite (0.1, 0.4 or 1mM) for 4h (n = 6).

Comment 3:

“(2) The details of the endogenous generation of HNO are still unclear. Whether and how lipoate promote HNO generation in the cells?”

Response:

We apologize if the descriptions of our findings and model were unclear. It has been shown that dihydrolipoate (DHLLA) reduces nitrosothiols like GSNO to GSH and HNO via the intermediate formation of HS-DHLLA-SNO (PMID: 7733655, PMID: 16277524). We propose that the NO production of M1 macrophages together with the substantial number of lipoate groups in the PDH complex are conditions ripe for HNO generation and signaling. The interaction of NO with reduced lipoate-bound to PDH-E2 subunit is likely to generate of HNO together with oxidized lipoate as a resulting product. We show that NO itself can interact with lipoate generating HNO, and we provide evidence that protein-bound lipoate (in our model PDHE2-) is able to significantly participate in this process. Such a mechanism is concordant with the lipophilicity of NO making it likely to accumulate at high concentrations in mitochondrial membranes, where it can encounter PDH and its ~108 lipoate groups, to facilitate HNO generation.

In Fig. 3 and S3, in a test tube reaction with limiting concentrations of GSH, we detect higher HNO production, as measured by accumulation of GSH-sulfinamide, when reduced lipoate is in the presence of NO (provided by the fast NO donor DEA/NO). Oxidized lipoate or NO alone yield little HNO. In addition, in Figure presented below for Reviewer 2, we have also measured direct HNO production, from test tube reactions, by using the florescent probe we describe in the manuscript.

Legend: Test tube reactions were carried out in HEPES (50mM, pH 7.4) buffer and fluorescent probe for HNO in the presence of DEA/NO with or without reduced-lipoate at indicated concentrations for 30 min at 37C. A mixture of buffer with the HNO donor AS was included as positive control.

In addition to providing this direct chemical evidence, we used ML226, an inhibitor of ABHD11, which protects the catalytic lipoyl domain of major mitochondrial alpha keto-acid dehydrogenases (Fig. 3e). ML226 reduces LPS+IFN γ -induced HNO generation, supporting the role of cellular lipoate in endogenous HNO generation.

We posit that the HNO resulting from exposure of NO to PDHE2-, will promptly target DLD due to proximity, react with its thiols and form a sulfinamide modification. Our data suggest Cys⁴⁸⁴ as a target. This residue has not been previously implicated in regulation of PDH, but our modeling suggested it might control DLD dimerization. A mutagenesis study, (now included in the revised manuscript at the highlighted lines 350-364 and 453-455), shows that mutation of that residue affects both dimerization and DLD activity.

It should be noted that in the mechanism we describe, there is no heme or Fe-S cluster involved since none of the proteins belonging to the PDH complex ligates these metal cofactors. Therefore, there is no HNO “binding” but rather direct reaction with target thiol groups.

Comment 4:

“(3) Since the addition of lipoyl modifications to substrates has been best characterized via two independent pathways (please refer to PMID: 29169048), either environmentally-obtained lipoic acid or de novo lipoic acid synthesis in mammals can support protein lipoylation. Therefore, whether the authors identified lipoylation of mitochondrial proteins including PDH complex by adding lipoic acid (200 μ M) as well as modulating the de novo pathway in cell culture? (please refer to PMID: 33174356, PMID: 34581824)”

Response:

The reviewer brings up possibilities we have been considering. Increased PDH activity with addition of oxidized lipoate does make sense since, in theory, this can support enzyme activity via incorporation into new or existing E2. Alternatively, it may preserve PDH activity by scavenging HNO, thus preventing HNO-induced PDH inhibition. Therefore, both pathways seem to be possible explanations for our finding in Fig. 1c and S2f.

Regarding scavenging of HNO by lipoate, HNO itself should not react with oxidized lipoate, only reduced lipoate. Reaction of HNO with reduced lipoate should form an N-hydroxysulfenamide intermediate that will be attacked by the second thiol to ultimately form hydroxylamine and oxidized lipoate. This is likely the only reaction route

given the close proximity of the two thiols of lipoate. Therefore, reduced lipoate could scavenge HNO directly resulting in less targeting of DLD. However, this reaction scheme would require reduction of the lipoate intracellularly first and then the now reduced lipoate would have to have access to the site of HNO generation for it to effectively prevent targeting of DLD. Although this is possible, we do not observe increased dihydrolipoate in treated cells, nor do we see substantial changes in GSSG/GSH levels, arguing that lipoate is not working as redox regulator per se (now Fig S2g).

An alternative explanation might be that exogenous lipoate is incorporated into newly generated E2. Lipoamide is covalently attached to specific lysine residues of target proteins through a complex series of enzymatic reactions, involving octanoyltransferase (*Lipt2*) and lipoyl synthase (*Lias*), both of which appear to be expressed in murine macrophages. Thus, there are two distinct possibilities for the source of lipoylation, *de novo* generation or incorporation via the lipoate salvage pathways. In our experiments, the effects are due to added lipoate making effects on *de novo* synthesis unlikely.

In bacteria, the salvage pathway involves conversion of free available lipoic acid to an adenylate intermediate (lipoyl AMP) by the action of lipoic acid ligase, followed by the transfer of the lipoic acid moiety directly onto target enzymes. The existence of this salvage pathway in eukaryotic cells is controversial and the required enzymes, such as lipoic acid ligase, have not been definitively identified in eukaryotic cells. However, studies have suggested that eukaryotic cells can salvage lipoic acid from degraded proteins via the mitochondrial enzyme medium chain acyl-CoA synthetase (ACSM1), which would utilize GTP to activate lipoic acid (PMID: 11470804 and PMID: 11382754). These reactions have not been studied in macrophages, but our previous dataset (PMID: 32019928) and unpublished findings suggest that these cells do not express *Ascm1*, but do have substantial levels of *Lipt1*. Whether murine *Lipt1* could possess lipoyltransferase activity as well as octanoyltransferase activity or not is unknown, but we have observed increased lipoyl-GMP when we feed cells exogenous lipoate (see new Fig S2g). The presence of activated lipoate is consistent with a salvage reaction in macrophages, but further study would be required to fully delineate this pathway.

The finding of lipoyl-GMP, together with evidence that exogenous lipoate is not working as a reducing agent, make it most likely that a salvage pathway is at work in macrophages and facilitating lipoyl- transfer to newly synthesized E2. We have clarified our test to reflect these conclusions from the new Fig. S2g-h at the highlighted lines 154-157.

Comment 5:

“(4) The abundant heme iron centers in mitochondria are available for the formation of HNO via the reaction between NO and H2S. Regarding the functions of modifications of PDH complex, whether and how PDH regulates iron oxidation state after E2/E3 modified by HNO (please refer to PMID: 27797441). .”

Response:

It is true that mitochondria are rich in heme proteins that may serve as site for HNO reactivity. We have shown HNO binds Fe(III) which may undergo reductive nitrosylation to Fe(II)NO (PMID: 12538052), which supports the conclusions of the publication cited by the reviewer. However, it is unclear what the reviewer is referring to regarding PDH regulation of iron oxidation after HNO-induced E2/E3 modification since PDH itself does not contain heme iron. It would seem possible that PDH may scavenge HNO preventing HNO reaction with cellular ferric heme centers, and thus preventing heme reductive nitrosylation to promote oxidized Fe(III) levels over reduced Fe(II).

Regarding HNO formation from H₂S and NO, this reaction requires the presence of a redox active metal center (e.g., Fe) since NO and H₂S would not be expected to react directly (NO is a radical and H₂S is not). In this case, HNO formation likely proceeds through the metal mediated formation of a nitrosothiol intermediate that can subsequently be reduced for HNO generation. Thus, if NO and H₂S levels rise in the presence of iron proteins, PDH activity could theoretically be affected through HNO formation and modification. Although this is possible, major cellular H₂S producers like CBS and CSE in macrophages are very poorly expressed, according to our previous expression data (PMID: 32019928) and unpublished RNAseq dataset, therefore we feel it is very unlikely that H₂S derived from these enzymatic activities may account for substantial HNO production in our system.

Comment 6:

“(5) The authors stated that depletion of cellular GSH with BSO, pushing the equilibrium of HNO chemistry towards sulfinamide (Fig. 2b). They should further use NAC (N-acetylcysteine) for GSH generation to reduce sulfinamide in the validation tests.

Response:

We thank the reviewer for suggestion. In our original manuscript, we showed (Fig. 5g) that with DLD treated in vitro with the HNO donor in the presence of GSH, that GSH levels control the degree of bioavailability of HNO and likely restrain the overall levels or detectability of irreversibly modified Cys⁴⁸⁴. However, to address this possibility further we have now treated BMDMs for 4h with NAC after 16h stimulation and confirmed increased intracellular GSH and a tendency towards restoration of DLD activity (figure for Reviewer 2 below, panels a-b). We also performed proteomics on NAC-treated samples using TCEP reduction followed by IAM-Peg2 tagging and enrichment of unmodified peptides. Indeed, we detected an increase in statistically significantly enriched peptides with higher abundance in NAC-treated cells, suggesting that, in accordance with our hypothesis, NAC is having a substantial effect in preventing irreversible modifications in macrophages (panel c). Moreover consistent with observed decreases in irreversible modifications, and in line with our BSO data, we find a NAC-induced increase in unmodified Cys⁴⁸⁴ peptide as compared to cells treated with LPS+IFN γ alone (panel d). These new data support the concept that reactivity of HNO with GSH intracellularly is important for scavenging the reduced NO species and preventing modification specifically of Cys⁴⁸⁴ (Cys⁶⁹ is not affected by NAC treatment, panel e) and further inhibition of protein activity.

Legend: WT BMDMs were stimulated with LPS+ IFN γ and treated with 5mM NAC for 4h after 16h stimulation. **a** Metabolites were assessed at ESI-LC/MS-MS to confirm NAC activity. **b** DLD dehydrogenase activity in

mitochondrial lysates. **C** volcano plot of IAM-PEG₂/TMT proteomic labelling after TCEP reduction. Carbamidomethyl (CAM) group depicts unmodified peptides. **d-e** normalized abundance of CAM-Cys⁴⁸⁴ and CAM-Cys⁶⁹ containing peptides.

Minor Comments

Comment:

“(1) The reference format should be corrected by using abbreviation of the journal names.”

Response:

We have used the *Nature* format for references in Endnote. We are at the discretion of the editor and the editorial management team regarding modifications concerning this aspect.

Comment:

“(2) Metabolomics also support the idea that BCKDH, OGDH and GLDC are targeted in a NO-dependent manner (Fig. S9d). I suggest that they further perform stable isotope tracing analysis for validation of BCKDH, OGDH and GLDC activities.”

Response:

We appreciate the suggestion made by the reviewer. Seim et al have shown in a previous report that activated macrophages display a decreased flux through OGDC, likely caused by dynamic changes of lipoylation status (PMID: 32259027). They later revealed that reductive nitrogen species can cause substantial inhibition of PDHC and OGDC (PMID: 36266351). Rather than repeating these experiments here, we have addressed this reviewer’s comment by focusing on BCKDH and GLDC.

We carried out [U-¹³C₆] Leucine tracing and quantified labeling on downstream intermediates. As expected, we observed equal proximal labeling and decreased ¹³C incorporation into intermediates downstream of BCKDH only in stimulated WT BMDMs. These results are now reported in new Fig 2a-c and S2c-e. These data clearly indicate that NOS2-derived NO drives cells to subvert BCAA oxidation by targeting DLD subunit. To our knowledge, this is the first report of regulation of leucine carbon fueling by NO in primary macrophages during stimulation, so we have now included a portion of these data in the main figures and in the revised manuscript at the highlighted lines 137-148 and 468-473.

In response to this comment, we also looked at glycine cleavage in the mitochondria of BMDMs. Although in our hands we detected >50% intracellular labeled glycine, suggesting uptake, the chemical complexity and stability issues with many of the metabolites in this pathway made measurements of their incorporation of the heavy carbon technically challenging, and therefore, we were not confident in the data. We have included some results for the Reviewer below. These data did suggest that GCS-derived formate from carbon 2 of glycine is apparently not incorporated into serine or methionine (panel a-b), suggesting that other major routes or extracellular uptake from the media are engaged for the synthesis and homeostasis of these. Small amounts of ¹³C were observed in adenine and N-formylmethionine, with some tendency for NOS2-dependent incorporation during stimulation (panel c). In summary, it appears this pathway is not highly active in macrophages under the conditions we utilized (panel d), so we have chosen not to add these data to the manuscript.

Regardless, we are confident that given our new data, distinct NOS2-dependent effects have now been documented on PDH, OGDH, and BCKDH, that even without documented effects on GCS, our conclusions that targeting of DLD by NO affects multiple DLD-dependent pathways are sound.

Legend: **a** schematic illustrating Glycine Cleavage System (GCS)-mediated oxidation of glycine and labeled intermediates resulting from [U-¹³C₂] glycine. **b** WT and *Nos2*^{-/-} BMDMs were activated for 16h with LPS + IFN γ and cultured with U¹³C-glycine for additional 4h. Bars show total % abundance of labeled M1 isotopologue of Methionine and Serine. **c** fractional labeling and % abundance of labeled M1 isotopologue respectively of adenine and N-formyl methionine. **d** % abundance of labeled M2 isotopologues of GSH and serine, which represent glycine carbon utilization that does not involve oxidation through GCS.

Reviewer 3.

Comment 1:

"In the abstract and in various places throughout the paper the authors refer to HNO directly targeting the PDH complex. This is misleading. More appropriate it to describe this as a facile chemical reaction, likely due to the proximity of generation."

Response:

We thank the reviewer for the suggestion to better clarify the mechanism we propose for PDH inhibition. Indeed we have now emphasized repeatedly throughout the manuscript the importance of proximity of DLD to HNO generated at E2 and the overall chemistry of HNO attacking thiols close to the site of generation.

Comment 2:

"A thorough grammatical review by the authors is recommended?"

Response:

We apologize for the presentation issues and have revisited the text throughout.

Comment 3:

"The authors state (line 43) this HNO formation is "peculiar to the lipoate rich PDH complex". I agree but this very description limits the breadth of the findings. It is a very good piece of mechanistic work on this specific reaction. "

Response:

The reviewer raises an intriguing point. Although the mechanism we describe here focuses on understanding PDH inhibition, highlighting an intramolecular phenomenon involving NO and HNO, we feel the data likely have substantial impact outside of PDH. For example, as noted to reviewer 2, minor comment (2), the inhibition of DLD affects several metabolic pathways. In addition, our description of lipoate-facilitated HNO production means that other targets nearby may be accessible. For example, our proteomics analysis revealed a signature associated with cell shape and cytoskeletal activities (Fig. 4d-e), consistent with our previous reports of NO-dependent morphological changes during M1 polarization (PMID: 31683257). Moreover, it is possible that other sites of generation can exist facilitated by as yet undescribed thiol rich sites, likely close to the cell membrane. With our description of in situ, physiological production of HNO, all these possibilities come into play and should be investigated.

Comment 4:

"The "rewiring" concept is an overreach (line 62 and throughout). For example, how is this different from when a cell utilizes fatty acids under limiting glucose? It is an alternate pathway. Rewiring would involve more than just the use of an alternate pathway to generate energy. ."

Response:

Herein, we used the term re-wiring because a specific pathway is blocked by modification, as opposed to adaptation to alternative fuel sources or pathways. That said, the reviewer's point is well taken, and we have revisited the use of this term in the paper.

Comment 5:

“Line 63 mentions “suppression” of the PDH complex. This is wrong word to describe what is happening and I think leads the reader astray. The PDH complex is inhibited and thus pyruvate concentrations increase with the attendant regulation by pyruvate taking over”.

Response:

We appreciate the reviewer’s suggestion and we have now updated the text to more accurately refer to “inhibition” instead of “suppression” throughout the manuscript.

Comment 6:

“In lines 79-91, the authors summarize their work and others concerning how HNO might form and aspects of reaction rates with other molecules but they conclude with speculation on HNO as a signaling agent. Nothing in the literature as yet outlines the key aspects of signaling for HNO (e.g. spatial and temporal control) but more importantly, this paper is not about signaling and does not add to arguments in favor of HNO signaling but instead is focused on the formation of HNO from lipoate in the E2 subunit and subsequent reactions. Again, referring to the reaction as “targeting” (line 89) is misleading. Same on line 96. Better wording would be ‘reacts with’. It is more indicative of what is taking place. “

Response:

We regret that wording was not completely clear and accurate at times. Signaling by HNO has been hypothesized for three decades in the context as a small molecule bioregulator. All such species are reactive such that spatial and temporal control is an inherent aspect in physiology. The reactivity of HNO is particularly extensive, for example compared to NO and H₂O₂, and thus identification as an endogenous signaling agents has been quite challenging. In culmination of extensive chemical, biochemical and pharmacological studies in many research labs, we have been able to for the first time demonstrate a pathway of biosynthesis of HNO proximal to a critical target.

Comment 7:

“Beginning with line 107 the authors describe lipoate supplementation experiments. This is a complicated experiment. Scavenging of lipoate is pretty well worked out in bacteria so if we assume the same machinery exists in eukaryotic cells then, at a minimum, a lipoyltransferase would be required after activation of lipoate. Assuming these steps take place, how does PDH activity get revived? Is the complex disassembled and new E2 put in? Is the entire complex tagged for degradation and new PDH made? If the latter, why do you need exogenous lipoate? “

Response:

We would ask that the reviewer consider our response to Reviewer 2, comment 4. In addition, to answer the reviewer’s comment about why exogenous lipoate is needed upon new protein synthesis, others have showed (PMID: 32259027 and PMID: 29784770) that lipoylation is decreased during stimulation, and that this is due to Fe-S cluster biogenesis enzymes being targeted first by NO, and then transcriptionally repressed. LIAS, lipoate synthase, is a Fe-S protein therefore its downregulation during stimulation will interfere with lipoylation from de novo synthesis onto new E2. We think that exogenous lipoate may be shuttled via a salvage pathway and could

then contribute to lipoylation of new E2 and amelioration of protein activity (We have clarified our test to reflect these conclusions from the new Fig. S2g-h at the highlighted lines 154-157).

Comment 8 :

“The proteomic analysis (beginning on line 182) ???is this endogenous or with AS?? indicates rather broad reactivity of HNO. There is some focus on mitochondrial proteins but others not. The arguments about localization and reactivity seem weaker in light of these results.”

Response:

We understand the reviewer’s concern. Our work has addressed a “mitochondrial enriched” proteome, obtained as specified in the methods section. As stated in response to reviewer’s comment n3, together with possible metabolic targets of HNO, we found that broader cellular processes may be affected by this reactive specie. Although we characterize an important source of HNO, it is likely that there are other sources of endogenous HNO in addition to PDH. That said, we are operating under the well-developed assumption that due to the highly reactive nature of HNO, its sphere of influence should be limited. Given that our data are the first to characterize in detail a cellular HNO generator, little is known regarding the possible diffusion of HNO, or the efficiency of its intracellular scavenging.

Perhaps additional extensive analysis of lipoate-dependent targets will be informative here. However, given the highly reactive nature of HNO, we find it likely that this characteristic is responsible, in part, for the targeting of nearby DLD. Regardless, we have re-evaluated our claim of the localization of generation argument throughout the text.

Comment 9:

“The in vivo approximation experiments to find DLD Cys modifications (line 248 and following) are a concern. The sensitivity seems high enough for detection.”

Response:

We understand that a limitation of this work is the lack of direct measurement of these cysteine modifications in macrophages. Although click chemistry approaches have been successfully applied for the detection of global sulfenic and sulfinic cysteine modifications, this has been shown essentially just as a proof of principle on H₂O₂ treated cancer cell lines (PMID: 25175731, PMID: 30177848). Studies of targeted proteomics for quantification of sulfenic and sulfinic acid-cysteines in primary cells have not been performed to our knowledge. The closest study we refer to in our manuscript is a study that assesses global cysteine reversible oxidations in T cells activated *ex vivo* where Cys⁴⁸⁴ of DLD appears to be a target, validating our hypothesis of it being a physiologically relevant residue for immune cell physiology.

Comment 10:

“The in silico results reported on line 283 and following beg for experimental verification and are not compelling without such experiments. “

Response:

We agree with the reviewer that validating structural data can provide insight. To that end, we have now examined the role of Cys⁴⁸⁴ in DLD activity by expressing WT DLD in parallel with C484A and C484W mutants. We assessed

expression, dimerization and activity. These data now presented in Fig 6c-f show that mutation of Cys⁴⁸⁴ did not decrease the expression of the DLD protein, but that dehydrogenase activity was almost completely ablated in the mutants, supporting a fundamental role for this cysteine. Moreover, in native gels, we also observed that diaphorase activity was substantially reduced, apparently due to the inability of the mutated protein to generate effective dimers. Interestingly, while dehydrogenase activity was comparably reduced when Cys⁴⁸⁴ was substituted with either alanine or tryptophan, the alanine mutation partially retained some dimerization capacity as compared to tryptophan mutant. This is a potentially compelling observation that we are evaluating for further study. These new findings of the importance of Cys⁴⁸⁴ provide important insight into the understanding of this enzyme. We have now included part of these data in the main figures and in the revised manuscript at the highlighted lines 350-364 and 453-455.

REVIEWERS' COMMENTS

Reviewer #1 (expert in macrophage immunometabolism):

The authors have addressed my concerns well. I recommend the publication of the manuscript in the current format in Nature Communications.

Reviewer #2 (expert in metabolomics):

The authors have addressed all of my comments. But there are some minor errors should be corrected.

(1) Line 895 in Page 26, "BMDM (~10⁷ cells per sample)" should be changed to "BMDM (~10⁷ cells per sample)".

(2) Line 918 in Page 27, "1 h at 37C" should be changed to "1 h at 37°C".

Reviewer #3 (expert in NO Signaling and gas sensing)
Absent.

Reviewer #4 (expert in nitric oxide signalling and nitroxyl), replacing Reviewer #3:

The manuscript provides novel evidence for the endogenous generation of nitroxyl (NHO), via the lipoate moieties of PDH-E2. The findings have implications for a role of HNO in cellular metabolism.

The manuscript has been improved via thoughtful revision and the inclusion of additional experiments to verify the key role of Cys484 in DLD activity, thus highlighting this cysteine residue as a likely target of HNO.

The authors have addressed all the concerns raised and I have no further comments.

RESPONSE TO REVIEWERS' COMMENTS

Reviewer #1 (expert in macrophage immunometabolism):

The authors have addressed my concerns well. I recommend the publication of the manuscript in the current format in Nature Communications.

We thank the reviewer for the positive comments.

Reviewer #2 (expert in metabolomics):

The authors have addressed all of my comments. But there are some minor errors should be corrected.

(1) Line 895 in Page 26, “BMDM (~10⁷ cells per sample)” should be changed to “BMDM (~10⁷ cells per sample)”.

(2) Line 918 in Page 27, “1 h at 37C” should be changed to “1 h at 37°C”.

We thank the reviewer for the positive comments. We apologize for these last oversights which now have been addressed.

Reviewer #4 (expert in nitric oxide signalling and nitroxyl), replacing Reviewer #3:

The manuscript provides novel evidence for the endogenous generation of nitroxyl (NHO), via the lipoate moieties of PDH-E2. The findings have implications for a role of HNO in cellular metabolism.

The manuscript has been improved via thoughtful revision and the inclusion of additional experiments to verify the key role of Cys484 in DLD activity, thus highlighting this cysteine residue as a likely target of HNO.

The authors have addressed all the concerns raised and I have no further comments.

We thank the reviewer for the positive comments.